# Cationic nanoparticle as an inhibitor of cell-free DNA-induced inflammation

Huiyi Liang[1], Bo Peng[1], Cong Dong[1], Lixin Liu[1], Jiaji Mao[2], Song Wei[3], Xinlu Wang[3], Hanshi Xu[4], Jun Shen [2], Hai-Quan Mao [1,5], Xiaohu Gao[1,6], Kam W. Leong [1,7] & Yongming Chen [1]

Cell-free DNA (cfDNA) released from damaged or dead cells can activate DNA sensors that exacerbate the pathogenesis of rheumatoid arthritis (RA). Here we show that ~40 nm cationic nanoparticles (cNP) can scavenge cfDNA derived from RA patients and inhibit the activation of primary synovial fluid monocytes and fibroblast-like synoviocytes. Using clinical scoring, micro-CT images, MRI, and histology, we show that intravenous injection of cNP into a CpG-induced mouse model or collagen-induced arthritis rat model can relieve RA symptoms including ankle and tissue swelling, and bone and cartilage damage. This culminates in the manifestation of partial mobility recovery of the treated rats in a rotational cage test. Mechanistic studies on intracellular trafficking and biodistribution of cNP, as well as measurement of cytokine expression in the joints and cfDNA levels in systemic circulation and inflamed joints also correlate with therapeutic outcomes. This work suggests a new direction of nanomedicine in treating inflammatory diseases.

[1] Center for Functional Biomaterials, School of Materials Science and Engineering, Key Laboratory for Polymeric Composite and Functional Materials of Ministry of Education, Sun Yat-sen University, 510275 Guangzhou, China. [2] Department of Radiology, Sun Yat-sen Memorial Hospital, Sun Yat-sen University, 510120 Guangzhou, China. [3] General Hospital of Guangzhou Military Command of PLA, 510010 Guangzhou, China. [4] Department of Rheumatology, The First Affiliated Hospital, Sun Yat-sen University, 510080 Guangzhou, China. [5] Department of Materials Science and Engineering, and Institute for NanoBioTechnology, Johns Hopkins University, Baltimore, MD 21218, USA. [6] Department of Bioengineering, University of Washington, Seattle, WA 98195, USA. [7] Department of Biomedical Engineering, Columbia University, New York, NY 10027, USA. Correspondence and requests for materials should be addressed to L.L. (email: liulixin@mail.sysu.edu.cn) or to J.S. (email: shenjun@mail.sysu.edu.cn) or to K.W.L. (email: kam.leong@columbia.edu) or to Y.C. (email: chenym35@mail.sysu.edu.cn)

Rheumatoid arthritis (RA) afflicts approximately 1% of population worldwide. Its symptoms include swollen and deformed joints due to inflammatory damage of the bone and cartilage[1]. Severe cases could lead to systemic failure including cardiovascular, pulmonary, psychological, and skeletal disorders. Current drug therapy relies on glucocorticoids and non-steroidal anti-inflammatory drugs[2]. Synthetic anti-rheumatic drugs, such as methotrexate show strong side effects like teratogenicity and hepatotoxicity[3,4]. Biologics such as anti-TNF-α that acts by inhibiting inflammatory functions may cause a high risk of infection, like tuberculosis, due to a weakened immune system[5]. Moreover, biologic drug is too expensive for patients in developing countries.

It is still a great challenge to treat RA, especially for its complicated pathogenesis[2]. Recent studies suggested that cell-free DNA (cfDNA), played a critical role in RA development. The origin of cfDNA comes from degradation of DNA released from apoptotic or dead cells, nuclei expelled from erythroid precursors, mitochondrial DNA, or neutrophil extracellular traps (NETs)[6,7]. Clinical study showed that cfDNA level in serum of patients was elevated[8,9]. The cfDNA content in the synovial fluid of patients is even thousands of times higher than that in the plasma, serving as a biomarker for RA diagnosis[10]. cfDNA eliciting immunity response in vivo through varied ways has been recognized[11]. The immune complexes formed by cfDNA and autoantibody, as well as proteins like HMGB1 and LL37 may be internalized into immune cells and activate the Toll-like receptors (TLR) to secrete inflammatory cytokines to cause tissue inflammation[12–14]. The immunocomplexes of cfDNA can promote B cells to proliferate and differentiate in RA patients, implying that cfDNA elicits not only innate immunity but also adaptive immunity of the patients[15,16]. Furthermore, several other studies showed a profound relationship between cfDNA and RA development. When excessive DNA was accumulated in the plasma of nuclease-deficient mice, it would lead to abnormally high-level cytokines that in turn caused joint inflammation and bone destruction[17,18]. These studies suggest that cfDNA may be a target for RA therapy. Recently, Sullenger et al. demonstrated that cationic polymers can be used to neutralize cfDNA to inhibit inflammation in an acute liver injury model[19] and in alleviating symptoms of systemic lupus erythematosus (SLE)[20].

Hypothesizing that blocking the TLR activation may reduce inflammation in RA, we propose the use of cationic polymers to scavenge damage-associated molecular patterns (DAMP) molecules as a new strategy to treat RA. In addition, we opt to focus on cationic nanoparticles (cNP) instead of soluble polycations because of a potentially higher nucleic acid (NA) scavenging capacity and a more favorable biodistribution in the inflamed joints (Fig. 1a). In the development of RA, inflammation induces angiogenesis to form a leaky vasculature in the inflamed joints[21,22]. We show that cNP composed of the diblock copolymer of poly(lactic-co-glycolic acid) (PLGA) and poly(2-(diethylamino)ethyl methacrylate) (PDMA) with a size around 40 nm have a high DNA-binding affinity, which in turn efficiently inhibit primary SFMC and FLS activation by cfDNA from RA patients (Fig. 1a). Intravenous injection of cNP into a CpG-induced mouse or collagen-induced arthritis (CIA) rat model[23] would relieve RA symptoms with respect to ankle and tissue swelling and reduce bone and cartilage damage. The cNP are effective in treating the CIA model at both the early stage, as well as the established stage of RA progression. Treatment with cNP also leads to partial mobility recovery in the CIA-induced RA rats. The favorable therapeutic outcome is corroborated with determation of intracellular trafficking, biodistribution, cfDNA levels in systemic circulation and inflamed joints, and cytokine levels in the joints.

## Results

### cNP shows high DNA binding and inhibition of inflammation.

cNP ca. 40 nm in diameter were prepared by self-assembly of PLGA-block-PDMA block copolymer, PLGA-b-PDMA$_{463}$ (see chemical structure in Supplementary Figure 1a). The particle has a structure of PLGA core and cationic PDMA shell. The homopolymer with a similar chain length to the PDMA shell of cNP, PDMA$_{470}$, was used for comparison. In PBS of pH 7.4, cNP and PDMA are polycationic materials with the zeta potential of +18 and +14 mV, respectively. The binding affinity of the cNP and PDMA with calf thymus DNA was evaluated by competitive binding with ethidium bromide (EtBr)[24]. In both PBS and 10% FBS, cNP showed a stronger binding ability than PDMA at polymer to DNA ratios lower than 0.6. The presence of serum did not change the DNA-binding efficiency of cNP at ratios higher than 0.6, but did weaken the binding at ratios below this value (Fig. 1b).

We next checked if the cationic materials could inhibit the NA-mediated activation of cell inflammation. Considering the cell toxicity of materials (Supplementary Fig. 2a) and binding affinity, we chose concentrations ≤50 μg/mL for the in vitro experiments. The read-out was given by TLR activation of the Ramos Blue$^{TM}$ reporter cells, which are constructed from human B lymphocyte. While the cationic materials alone would not activate the TLR pathway (Supplementary Fig. 2b), both cNP and PDMA at different concentrations (0.25, 2.5, and 25 μg/mL) inhibited the TLR activation in Ramos Blue$^{TM}$ cells by synthetic NA-based TLR agonists, such as TLR3 agonist polyinosinic-polycytidylic acid (poly (I:C)), a synthetic dsRNA, and human TLR9 agonist CpG-ODN2006 (Supplementary Fig. 3a). However, they could not inhibit the TLR activation by synthetic non-NA TLR agonists, such as the triacylated lipopeptide Pam3CSK4 for TLR2 and the imidazoquinoline compound R848 for TLR7, respectively. Similar inhibitory activity was observed in RAW264.7 cells, that cNP and PDMA could inhibit the immune stimulatory activity of poly (I:C) and CpG-ODN 1668 but not Pam3CSK4 and R848 (Supplementary Fig. 3b); neither cNP nor PDMA alone could up-regulate TNF-α level significantly (Supplementary Fig. 2c). These results indicate that these cationic materials are specific for NA-mediated activation of the TLRs. Furthermore, both cNP and PDMA could only block the stimulatory effect of a NA-based agonist ssRNA–lipid complexes (ssRNA40) on mouse TLR7, which is located in the endosomal membrane, but not the stimulatory effect of the non-NA-based agonist R848, which is positively rather than negatively charged. It is therefore logical from the viewpoint that the inhibition comes from the neutralizing effect of the cationic materials. However, there remains the question whether the cationic materials might bind to the TLR7 receptor and exert the inhibitory effect. To address this issue, we first treated RAW264.7 cells with ssRNA40 and the cationic materials, and then added the R848 one hour later. In the first step, the TLR7 stimulation would be blocked because of the binding of ssRNA40 by the cationic materials. However, the later addition of R848 should be able to activate the TLR7 pathway if the TLR7 receptor has not been pre-empted by the cationic materials. Indeed we observed significant up-regulation of the inflammatory cytokines stimulated by R848 (Supplementary Fig. 3c), confirming our hypothesis that the cationic materials inhibit the TLR activation by binding to the NAs.

We further studied the ability of the cationic materials in attenuating the DNA-mediated cell inflammation. cNP and PDMA could inhibit the CpG-ODN activation of Ramos Blue$^{TM}$ cells in a dose-dependent manner (Fig. 1c). At 25 μg/mL, cNP showed almost 10-fold higher inhibition efficiency compared with the PDMA, indicating that higher DNA-binding affinity correlates with better inhibition. The result was further confirmed by measuring the

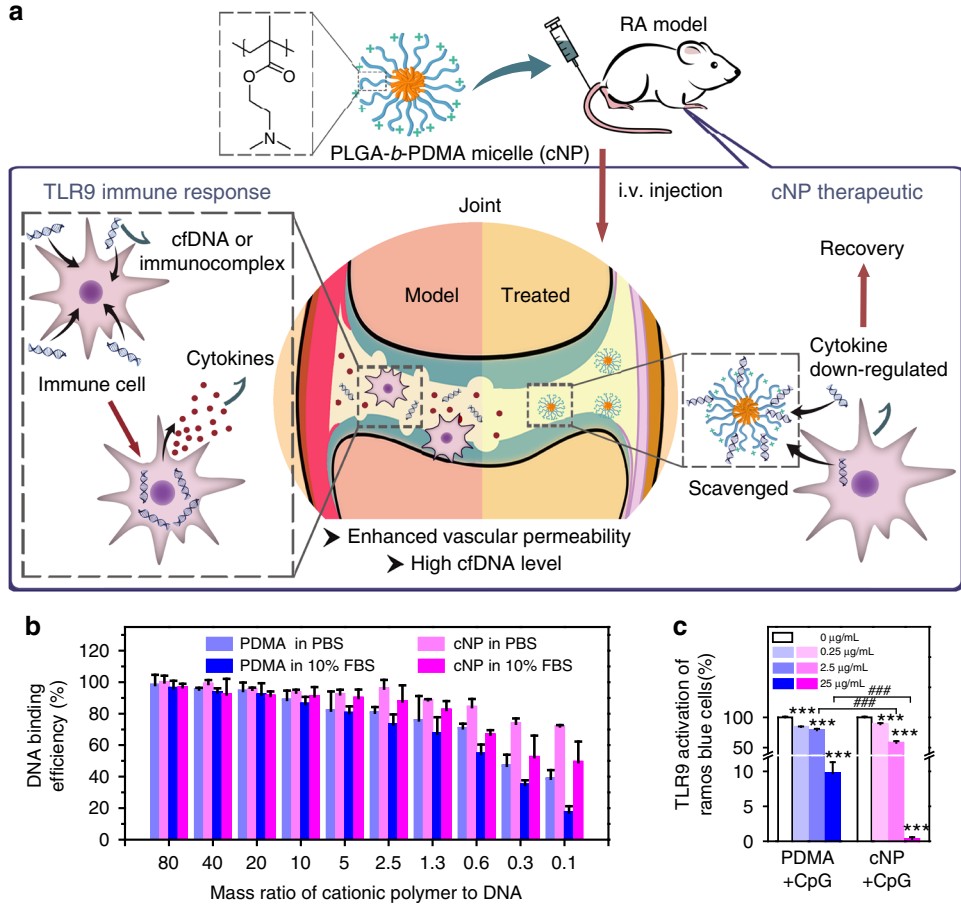

**Fig. 1** cNP shows high DNA-binding ability and efficient inhibition of TLR9 activation. **a** Mechanism of using cNP to scavenge cfDNA to inhibit inflammatory response for treating RA. cNP with PDMA corona is formed from PLGA-*b*-PDMA block copolymer self-assembly. In the RA animal models, damage-associated molecular patterns (DAMP) including cfDNA stimulate immune cells to release cytokines, which, in turn, recruit mononuclear cells to the synovium to damage the cartilage and bones. After intravenous injection of cNP to scavenge cfDNA, cytokines are down-regulated and symptoms are alleviated. **b** DNA-binding efficiency of PDMA and cNP in PBS and 10% FBS at 37 °C. **c** Inhibition of TLR activation in Ramos Blue™ cells using the QUANTI-Blue™ assay. Statistical significance was calculated by one-way ANOVA with the LSD post-test, ***$P < 0.001$ versus 0 μg/mL materials + CpG. ###$P < 0.001$ between two groups. In **b** and **c**, data are presented as the mean ± s.e.m

protein expression level in the downstream of the signal pathways by Western blotting. CpG 2006 activates immune cells through the MyD88-NFκB signaling pathway. Once TLR9 senses NAs, the adaptor protein MyD88 is activated to form a homo-oligomeric signaling complex, which in turn recruits TRAF6 to create an assembly platform that activates the TAK1 and IKK complexes, leading to MAPK and NFκB activation, respectively[25]. The results showed that both MyD88 and TRAF6 expression were down-regulated once CpG 2006 was neutralized by PDMA or cNP, with cNP showing a better performance attributed to its stronger inhibition of TLR9 activation (Supplementary Fig. 4a, b). Next, we explored the mechanism of inhibition possibly attributed to cellular uptake. One μM of Cy5-labeled CpG 1826 and 1 μg/mL of FITC-labeled polymers were both added into RAW264.7 cells for 12 h co-incubation, and their cellular uptake at 4, 8, 12 h was observed by confocal microscopy. Addition of PDMA or cNP did reduce the amount of CpG to be internalized (Supplementary Fig. 5a, b). This suggests that cationic polymers could neutralize extracellular inflammatory NAs, reduce their internalization and inhibit activation.

**cNP inhibits intracellular agonists to TLR9.** In the above experiments, cationic materials were added at the same time with

the CpG to the cell. In the non-idealized situation, the pathogenic NA would already be at the tissue and interact with immune cells when the scavengers reach them. To mimic this scenario, we first incubated the reporter cells with CpG and then the cells were washed thoroughly to remove any free extracellular CpG before the addition of the materials. Fig. 2a showed that, at the concentration of 25 μg/mL, cNP could decrease the activation of TLR9 to ~80%, whereas PDMA showed no effect. However, at dose of 50 μg/mL, both materials decreased the activation of TLR9 to 40%, implying that the TLR9-bound CpG could be captured competitively due to the strong binding affinities from the materials. This result was further confirmed by a different cell, the mouse macrophage cell line RAW264.7, at dose of 25 μg/mL, using the same protocol (Fig. 2b). Still, cNP showed better inhibition activity than that of PDMA. These results suggest that the cationic materials could enter the cells and capture the cfDNA from TLR9 in the endolysosomal compartment. To prove this, we conducted an intracellular trafficking study with cy5-labeled CpG 1826, selective to mouse TLR9, and the materials in RAW264.7 cells. Localization of CpG in the endolysosome suggests TLR9 recognition (Fig. 2c and Supplementary Fig. 6a). When FITC-labeled cationic materials were added and their intracellular location monitored over a 12-h period (Supplementary Fig. 6a for different time), the PDMA intensity increased but dispersed in

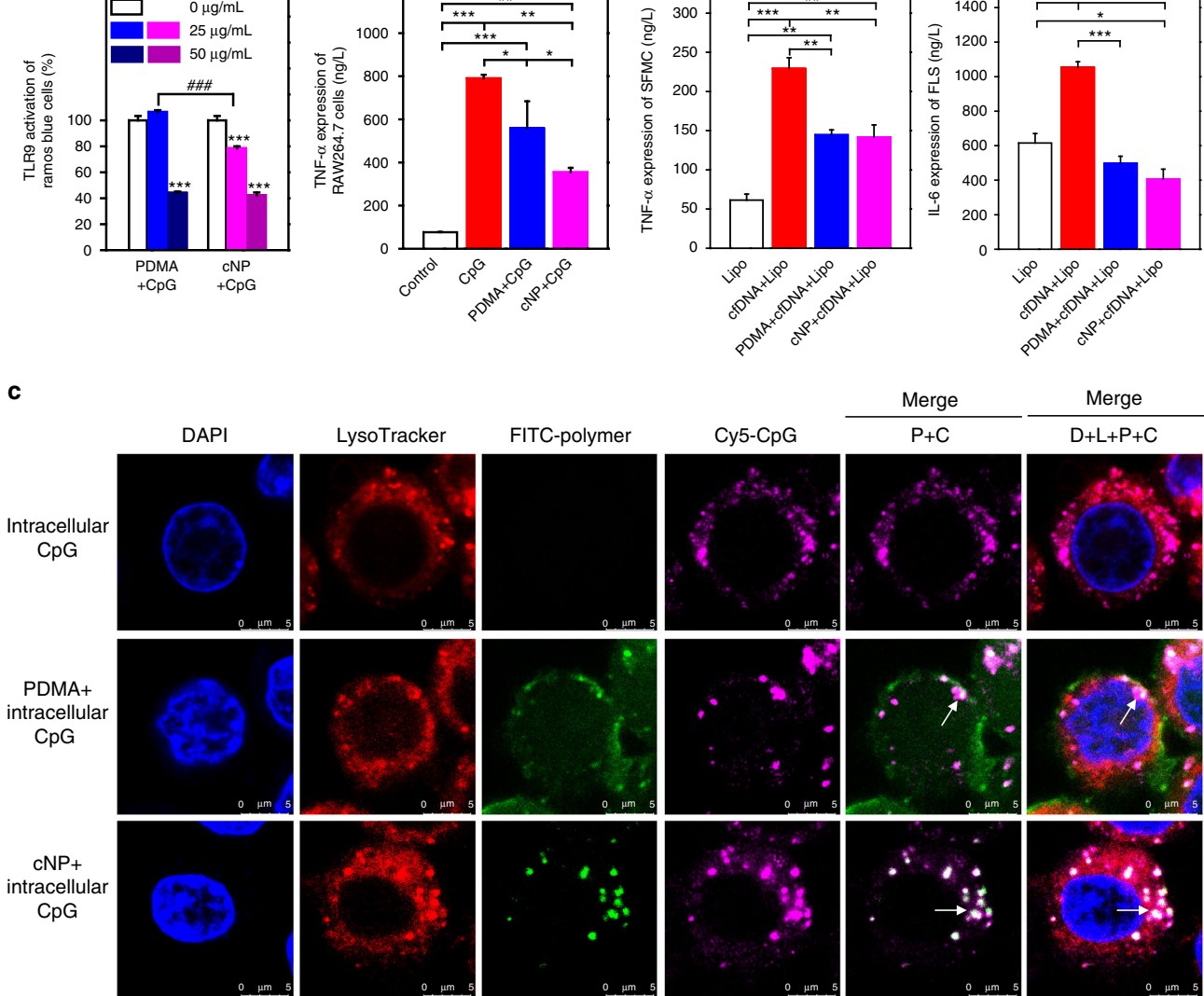

**Fig. 2** cNP better inhibits intracellular CpG to activate TLR9 via a stronger interaction with CpG. **a** Inhibition of TLR9 of Ramos Blue[TM] cells due to intracellular DNA scavenging by cationic materials. First, 1 μM CpG 2006 was added to Ramos Blue[TM] cells. After incubation for 4 h, cationic materials were added. Statistical significance was calculated by one-way ANOVA with the LSD post-test, ***$P < 0.001$ versus 0 μg/mL materials + CpG, ###$P <$ 0.001 between two groups. Data are presented as the mean ± s.e.m. **b** Cationic materials inhibited TNF-α expression in RAW264.7 cell. Firstly, RAW 264.7 cells were incubated at 37 °C for 4 h with 1 μM CpG 1826, then the medium was replaced with the medium containing cationic materials. After 24 h, CpG up-regulated TNF-α expression. After incubating with PDMA or cNP, TNF-α expression decreased. **c** Enlarged images show intracellular localization of intracellular CpG and cationic materials in RAW264.7 cells after 8 h incubation (Bar: 5 μm). Colocalization of CpG and cationic materials showed up as white spots, which are marked by the arrow. D indicates DAPI, L indicates LysoTracker, P indicates polymer, and C indicates CpG. **d** and **e** Cationic materials inhibited intracellular cfDNA to stimulate SFMC (**d**) and FLS (**e**). cfDNA and SFMC were collected from the synovial fluid of the same patient. In **b** and **d**, 25 μg/mL of the materials; in **e** 0.5 μg/mL of the materials. In **b**, **d**, and **e** statistical significance was calculated by one-way ANOVA with the LSD post-test, *0.01 < $P$ < 0.05, **0.001 < $P$ < 0.01, ***$P$ < 0.001. Data are presented as the mean ± s.e.m

the nucleus, endolysosome, and cytoplasm (Fig. 2c). In contrast, the cNP showed preferential localization in the endolysosome, where the CpG remained unchanged. Quantitative colocalization ratio of cNP in endolysosome at different time is much higher than that of PDMA (Supplementary Fig. 6b). This specific colocalization of cNP and CpG in the endolysomal compartment may help explain the therapeutic effect in the in vivo study.

We next investigated whether the cationic materials could also enter the primary SFMC and FLS from RA patients, to inhibit the inflammatory response by endogenous cfDNA also from the same patients. By transducing into the cells with Lipofectamine® 2000 (Lipo), cfDNA induced significantly more TNF-α and IL-6 from SFMC and FLS, respectively (Fig. 2d, e), while the materials alone

showed no significant response (Supplementary Fig. 2d, e). In the presence of either PDMA or cNP, the production of cytokines was inhibited obviously. The decrease of cytokines by cNP treatment was even more obvious than that of PDMA.

**cNP alleviates symptoms of CpG-induced acute arthritis.** One day after injection of 6 μg CpG into the articular cavity of mice, acute arthritis was set as manifested by swollen hindpaws. The animals were then treated with intravenous injection of cNP at a daily dose of 12.5 mg/kg for the following 7 days (Fig. 3a). Hindpaw swelling of the mice decreased gradually (Fig. 3b and Supplementary Fig. 7a). The diameter of the inflamed ankles

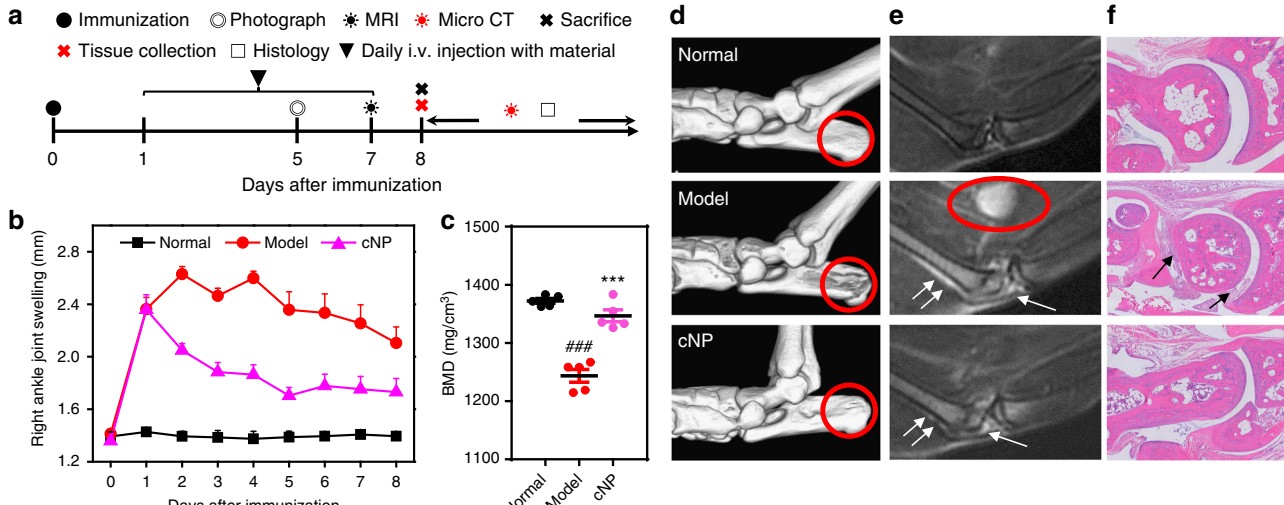

**Fig. 3** cNP improves the arthritic condition of BALB/c mice induced by CpG. **a** Experimental schedule of the mice study. Acute arthritis developed one day after articular injection with CpG. Then arthritis mice were injected i.v. daily with 12.5 mg/kg cNP from day 1 to day 7. Further study was performed after sacrificing mice at day 8. **b** Right ankle joint swelling of all mice was evaluated daily by measuring the diameter of the right ankle with a digital caliper ($n = 5$). **c** Analysis of BMD of mice ankle joint from Micro-CT data. Statistical significance was calculated by one-way ANOVA with the LSD post-test, ***$P < 0.001$ versus model group, ###$P < 0.001$ versus normal group. **d** Representative micro-CT images of the ankle joint of BALB/c arthritis mice after administration for 7 days with a resolution of 19 μm. 3D images were reconstructed using Inveon Research Workplace. Various levels of bone destruction in calcaneus are shown in red circles. **e** Representative T2-weighted MRI images of ankle joint of mice at day 7. The swollen popliteal lymph node (red circle), effusion in the joint (single arrow), and suprapatellar bursa (double arrow) appeared in the model, while treatment with cNP efficiently prevented these pathological changes. **f** Representative HE staining of ankle joints of mice after administration for 7 days (×200). Some inflammatory cell infiltration (single arrow) appeared in the articular cavity of model. In **b** and **c** data were presented as the mean ± s.e.m

decreased from 2.4 mm before injection to 1.7 mm at the end of 7 days. The micro-CT images show that the bone mineral density (BMD), a marker of bone loss, of the cNP-treated group (Fig. 3c) recovered from 1243 to 1346 mg/cm³, a level close to the normal mice. Bone destruction, as highlighted in the circled region in Fig. 3d, was undetectable after treatment. Percent bone volume and trabecular thickness also returned to those of the normal group (Supplementary Fig. 7b). The other parameters, such as bone surface area/bone volume, trabecular spacing and trabecular pattern factor, were decreased to the levels close to the normal control.

To assess synovitis, another common symptom of RA in addition to bone erosion, we applied MRI to evaluate the soft tissue disorder around the joints. The MRI images show the appearance of effusion in the knee and ankle joints, and suprapatellar bursa, as well as swelling of one popliteal lymph node in the model (Fig. 3e and Supplementary Fig. 7c). Whereas, after treatment, the effusion decreased and the popliteal lymph node shrunk to a size that was undetectable. The pathological assessment also indicates that the inflammatory cell infiltration at the synovium and articular cavity in the model was decreased after cNP treatment, with low histology scores close to the normal group (Fig. 3f and Supplementary Fig. 7d, e).

**cNP relieves symptoms of CIA rats in therapeutic treatment.** The above results show that cNP inhibited the inflammatory response of CpG-induced acute arthritis model by specifically blocking the TLR9 activation. However, in the CpG-induced arthritis model, T and B cells are not required in model developing, which is different from that of human RA[26]. To investigate whether the cationic materials can perform in a more mimetic model, we conducted the studies in the CIA model, whose inflammatory response is similar to that in human RA. In the CIA model, both T cells and B cells are abnormally activated and

the macrophages play critical roles in tissue damage during disease development. We evaluated the performance of cNP therapeutic treatment in both early and established stages of the model, aiming to assess the translational potential of cNP as a new strategy for treating RA (Fig. 4a). Firstly, the injection dose was optimized (Fig. 4b). At a dose of 25 mg/kg, all rats injected with PDMA died immediately, whereas cNP resulted in a death rate of 25%. At half the dose of 12.5 mg/kg, all rats in the cNP group survived while the PDMA group resulted in a death rate of 35.7%. Thus, the dose of 12.5 mg/kg for both cNP and PDMA and also 25 mg/kg for cNP were studied in four groups.

The CIA rat model is associated with symptoms of swelling and erythema in ankle and paw joints, as well as bone erosion due to inflammation in synovial cavity. For the early stage treatment, 13 days after the disease induction and when symptoms appeared, the animals were treated with a daily intravenous injection of the cationic materials for 15 days (Fig. 4a). A clinical scoring was used to evaluate the treatment effect[27]. The hindpaws of the model groups swelled gradually and reached a maximum of 2.2 mL at day 21, while the forepaws swelled more gradually (Fig. 4c). At a dose of 12.5 mg/kg, both the cNP and PDMA groups showed reduced swelling in hindpaws and forepaws (Fig. 4c and Supplementary Fig. 8a). Beyond day 20, the cNP groups showed a clearer inhibition of the hindpaw swelling than the PDMA group (Fig. 4c). At day 28, the hindpaw joint swelling in the cNP group decreased by 50%, while the PDMA group only decreased by 25%. At the higher dose of 25 mg/kg for cNP, the animals showed improved clinical scores in both of their hindpaws and forepaws. Thus, for the cNP groups, the 25 mg/kg dose was more effective than the 12.5 mg/kg dose.

The micro-CT analysis of inflamed ankle joints at day 29 showed a serious bone erosion in the model group, with the BMD dropping from 1564 to 1246 mg/cm³ (Fig. 5a, b). Treatment with 12.5 mg/kg of PDMA between days 13 and 28 saw a small improvement with the BMD to 1309 mg/cm³. On the contrary,

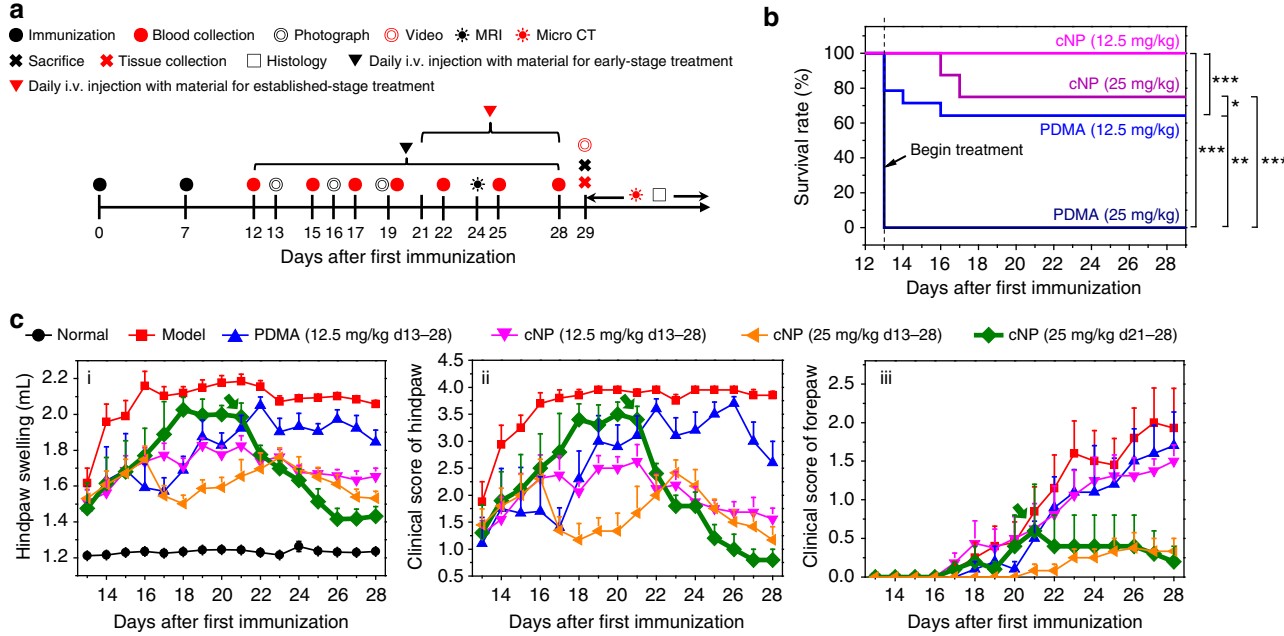

**Fig. 4** cNP better alleviates the swelling of joints in the CIA model in both early and established therapeutic treatment. **a** Experimental schedule of the CIA model study. After immunization with collagen type II and Freund's adjuvant twice (at day 0 and day 7), the CIA model was developed. Then CIA rats were i.v. injected with cationic materials from day 13 to day 28 for the early-stage treatment, or i.v. injected with cNP from day 21 to day 28 for the established-stage treatment. Photos and blood were taken regularly during therapy. Further study was performed after the rats were sacrificed at day 29. **b** Survival curve of different treatment groups from day 13 to day 28 after disease progression. Initial number of rats: $n = 5$ for PDMA, 25 mg/kg; $n = 14$ for PDMA, 12.5 mg/kg; $n = 16$ for cNP, 25 mg/kg; and $n = 13$ for cNP, 12.5 mg/kg. Statistical significance was calculated by a log-rank (Mantel–Cox) test, *$0.01 < P < 0.05$, **$0.001 < P < 0.01$, ***$P < 0.001$. **c** Hindpaw swelling (**ci**), clinical score of hindpaw (**cii**), and forepaw (**ciii**) of rats treated in the early stage or the established stage of RA progression (thick green curve). (**ci**) Average hindpaw swelling was determined daily using a plethysmometer, and (**cii** and **ciii**) the average clinical score of the hindpaws and forepaws of all rats was evaluated. $n = 10$ for model, PDMA (12.5 mg/kg), cNP (12.5 mg/kg), and cNP (25 mg/kg) groups. $n = 5$ for the normal group and cNP groups (25 mg/kg d21–d28). Thick green curve is highlighted for the established-stage treatment with cNP (25 mg/kg). Data were presented as the mean ± s.e.m

the BMD of the cNP groups at low and high doses were 1391 and 1441 mg/cm³, respectively, closer to the normal group (Fig. 5b) and demonstrating clear inhibition of bone erosion. The trabecular parameters also confirmed that cNP group showed better anti-inflammatory effect than the PDMA group and that the higher dose was more effective (Supplementary Fig. 8b). In addition, MRI images taken at day 24 showed a large amount of joint effusion and severe soft tissue swelling in the ankle and foot of model rats (light area in Supplementary Fig. 8c). The swelling was decreased by treatment with 25 mg/kg of cNP. Histological analyses of various joints (Fig. 5c for knee joints and Supplementary Fig. 8d for other joints) indicated a severe bone and cartilage damage accompanied by a large amount of infiltration of mononuclear cells in the model rats. In the low-dose treatment groups, although there was still evidence of inflammatory cells in the synovium in all groups, the cNP-treated rats showed reduced bone and cartilage damage compared with the rats in the model group and the PDMA group. In the high-dose cNP group, the protection from cartilage and bone damage was more obvious compared with other groups when treatment was initiated at the early stage, and could be seen even with treatment initiated at the established stage compared with the model group.

We next evaluated the therapeutic effect of cNP for the model rats at an established stage. When the paw swelling peaked at day 21, a daily dose of cNP at 25 mg/kg was injected intravenously for 7 days (Fig. 4a). Surprisingly, the swollen hindpaws and forepaws both shrunk considerably and even recovered to the same extent as the early treatment (Fig. 4c, highlighted by thick green curve), implying that the inflammation was effectively controlled.

Furthermore, the erosive bone damage was arrested, as evident by the BMD value of 1442 mg/cm³ (Fig. 5b) and the trabecular parameters (Supplementary Fig. 8b). The histological study also showed that, compared with the model group, the synovium and other soft tissues were restored even in the treatment at the established stage, and histology scores of knee joints, ankle joints, digital joints and wrist joints were obviously lower after treated by cNP (Fig. 5c and Supplementary Fig. 8d, e).

Advanced RA patients often lose their agility and it is very challenging to restore even the basic mobility. We finally evaluated the mobility of the model and treated CIA rats in a rotational cage test (Fig. 5d, e and Supplementary Movie 1). We chose the most relevant cNP group, 25 mg/kg dose in the treatment of the rats at the established stage, for this behavioral assay. In contrast to the untreated CIA rats that could barely move, the cNP group could manage an average rotational speed of 3.5 rpm, while the normal group could stand the rotational speed up to 7 rpm.

To understand the different effects of the cationic materials and the mechanism of the therapeutic effects, we measured the biodistribution of the cationic materials, cfDNA levels in circulation and the synovial joints, and cytokine profiles in the joints. We studied the biodistribution of the cationic materials in the joints with time using in vivo near-infrared fluorescence (NIRF) imaging (Supplementary Fig. 9a), and determined their biodistribution in the main organs and joints using ex vivo NIRF imaging (Fig. 6a). In normal rats, the biodistribution patterns of the PDMA and cNP groups differed only in the lung, liver, kidney, and spleen but not in the joints (Supplementary Fig. 9b). In the model group, the PDMA accumulated mainly in the lung,

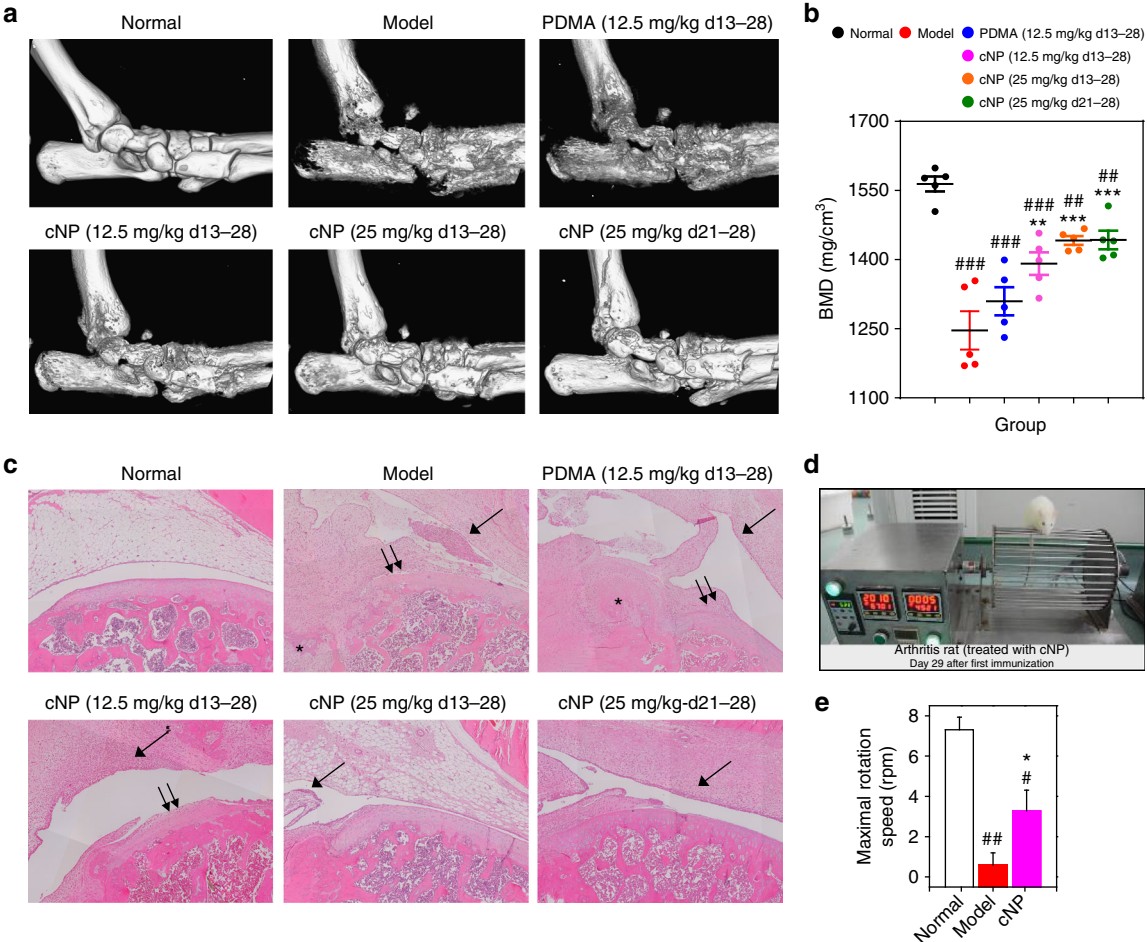

**Fig. 5** cNP shows better bone protection and alleviation in inflammatory cell infiltration for therapeutic treatment of CIA model. **a** Representative micro-CT images of the ankle joints with resolution of 19 μm. 3D images were reconstructed using Inveon Research Workplace. **b** Analysis of BMD of the rat ankle joints. Statistical significance was calculated by one-way ANOVA with the LSD post-test, \*\*0.001 < *P* < 0.01, \*\*\**P* < 0.001 versus the model group. ##0.001 < *P* < 0.01, ###*P* < 0.001 versus the normal group. **c** Representative HE staining of knee joints at day 29 after administration for 15 days (×200). Inflammatory cell infiltration in synovium (single arrow), cartilage destruction (double arrow), and bone destruction (\*) were clearly evident in model and PDMA groups, while treatment of cNP efficiently prevented these pathological changes. **d** Rotation cage test of CIA rats at day 29 after first immunization (Video was supplied as Supplemental Movie 1). **e** Maximal rotational speed at which rats could withstand during the rotation test at day 29 (*n* = 3). The cNP group was injected with 25 mg/kg cNP in period of day 21–28. Statistical significance was calculated by one-way ANOVA with the LSD post-test, \* 0.01 < *P* < 0.05 versus model group, #0.01 < *P* < 0.05, ##0.001 < *P* < 0.01 versus the normal group. In **b** and **e**, data were presented as the mean ± s.e.m

liver, and kidneys at 4 h, and relatively less in knees and paws. In contrast, cNP had a much higher accumulation in the knees and paws but much less in the other organs (Supplementary Fig. 9c). Moreover, cNP showed a longer retention in the inflamed joints (Supplementary Fig. 9a). The cfDNA level in circulation began to rise in model rats from day 12 till the end of the experiment (Fig. 6b). PDMA treatment had little effect on the systemic cfDNA level. In contrast, cNP treatment brought the systemic cfDNA down to a level mirroring the normal rats throughout the experiment. At the end of the experiment when the synovial cfDNA was measured in the sacrificed animals, the cNP group still showed a significantly lower level than the model and the PDMA groups, although significantly elevated compared with the normal rats (Fig. 6c).

We also measured the mRNA level of key inflammatory cytokines, including TNF-α, IFN-α, IL-6, and matrix metalloproteinase (MMP) 3 in the shin bones of CIA rats after administration for 15 days (Fig. 6d). Consistent with the lower cfDNA levels in the synovial joints, the cytokine and matrix metalloproteinase (MMP-3) levels related to the pattern recognition receptor (PRR) signaling pathways in the cNP group were all significantly reduced relative to the model group, and close to the normal group. In contrast, the PDMA group could show reduction only in IFN-α. The results have been confirmed by immunochemical analysis, the number of TNF-α, IL-6, MMP-3 immunoreactive cells were low in both cationic materials groups, among them cNP showed more efficiently inflammatory inhibition (Fig. 6e and Supplementary Figure 10).

## Discussion

Cationic polymers have been extensively applied to deliver NA for nonviral gene therapy for over two decades. Since 2009 Sullenger et al. have instead proposed to use cationic polymers to compete off aptamers[28] or remove proinflammatory NA for therapeutic purposes[19,29,30]. cfDNAs are NAs in nature that are recognized as DAMP or pattern-associated molecular patterns (PAMP) by PRR, such as the TLR in immune cells. Scavenging these DAMP molecules may limit inflammation. Using these nucleic acid-binding polymers (NABP) to remove the DAMP molecules, we had demonstrated that the scavenging approach

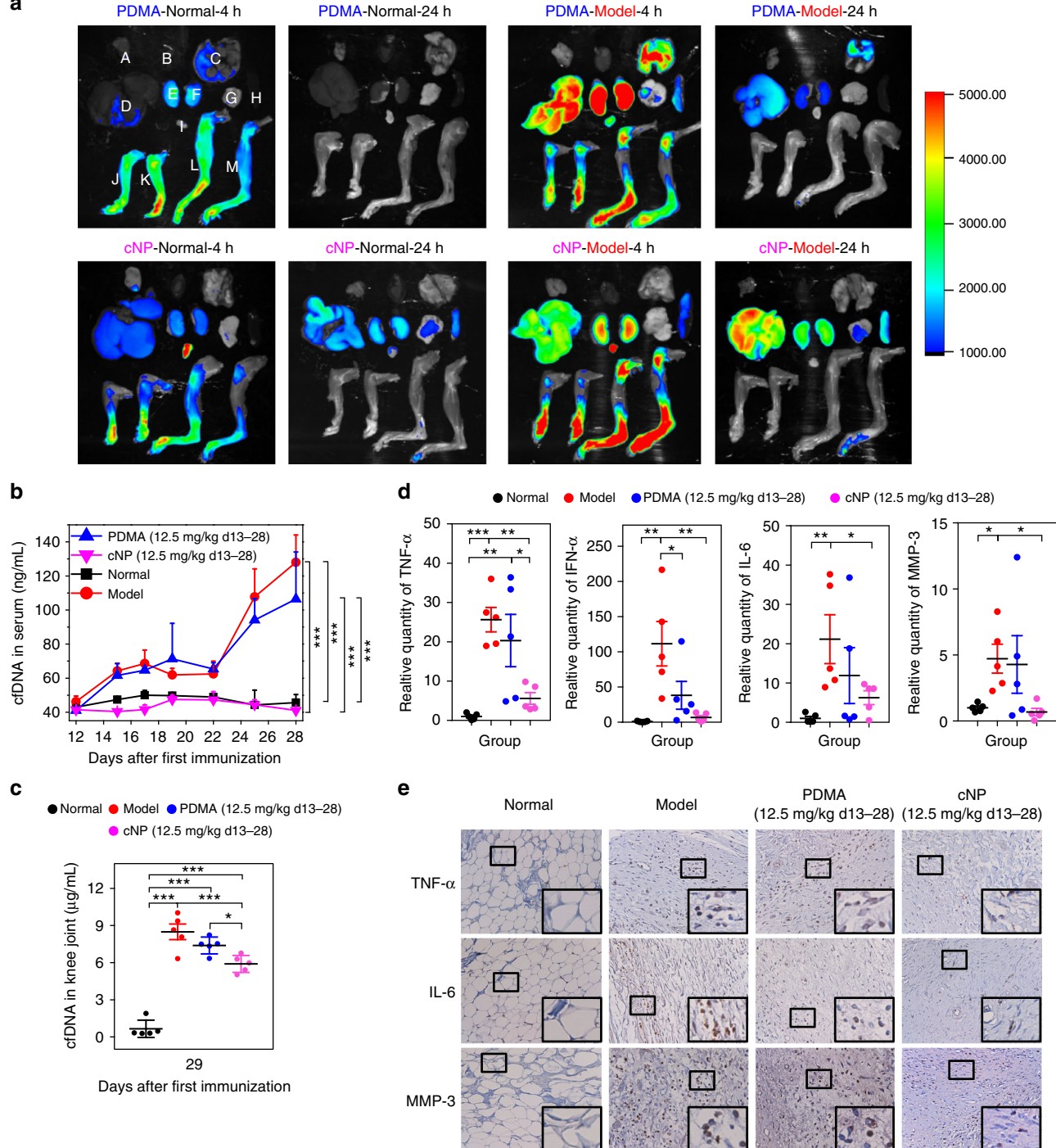

**Fig. 6** cNP accumulates more in inflamed joints, and more efficiently reduce cfDNA and proinflammatory cytokine expression in vivo than soluble polycations. **a** Ex vivo near-infrared fluorescence (NIRF) imaging of normal rats and CIA rats (clinical scoring 2) after i.v. injection of PDMA and cNP. Legend: A. thymus; B. heart; C. lung; D. liver; E/F. kidneys; G. pancreas; H. spleen; I. bladder; J/K. forelimbs; L/M. hindlimbs. **b** Changes of cfDNA concentration in the serum of rats treated with PDMA and cNP. **c** Concentration of cfDNA in knee joints after treatment with PDMA and cNP at day 29 after first immunization. **d** mRNA level of cytokines, TNF-α, IFN-α, and IL-6, as well as MMP-3 of tissue at day 29 after the first immunization. Relative mRNA expression of the cytokines and MMP-3 was normalized to GAPDH. In **b**, **c**, and **d**, data were presented as mean ± s.e.m. Statistical significance was calculated by one-way ANOVA with the LSD post-test, *$0.01 < P < 0.05$, **$0.001 < P < 0.01$, ***$P < 0.001$. **e** Detection of TNF-α, IL-6, and MMP-3 immunoreactive cells in synovial tissue of different groups. Synovial tissues of knee joints of different groups at day 29 were stained with anti-TNF-α, anti-IL-6, and anti-MMP-3 antibodies, respectively (×400). The inset showed the enlarged detail (×1000)

works in animal models of CpG-induced or RNA-induced acute liver injury[19], thrombosis[29], and SLE[20,30]. To reduce toxicity, we had immobilized the NABP on fibrous membrane to remove the DAMP molecules in a local[31] or ex vivo manner[32]. Motivated by the high NA-binding capacity of the immobilized configuration of NABP in these recent studies, we propose to use cNP as a DAMP scavenger, which has not been studied before, for RA treatment. Previously, Chauhan et al. observed that poly(amidoamine) dendrimers showed an anti-inflammatory activity toward arthritis, but no mechanism was discussed[33]. Dong et al. found that cationic dextran and PEI enhanced systemic concentrations of IL-12 and IFN-γ, and proposed that they in turn prevented neutrophil infiltration to the inflammatory sites in an adjuvant-induced arthritis (AIA) mouse model[34]. In these two studies, only soluble cations were used. Finally, Hayder et al. reported that azabisphosphonate (ABP)-capped dendrimers could ameliorate inflammation in a IL-1ra$^{-/-}$ murine model and a K/BxN serum transfer model to reduce damage in bone and cartilage[35]. The proposed mechanism was to inhibit inflammation activation via interactions with monocytes. Anionic in nature, this polymer would likely act through a different mechanism than what we proposed in this study.

Here, we established that the nanoparticulate format of NA-scavenger could achieve what the soluble counterpart could not in treating RA in rat models. We designed the PLGA-*b*-PDMA because PDMA by itself is a potent polycation used in nonviral gene delivery, and PLGA provides the hydrophobic core to form cNP. The binding of NA to cNP stems from charge interaction and we could optimize the NA-scavenging performance of the cNP by varying the molecular weight of PDMA. Moreover, the cNP would also likely interact with NETs and these interactions warrant further studies. The cNP should also bind to negatively charged proteins although its interaction with NA should be stronger than that with proteins. Therefore, in the presence of serum, cNP may still bind cfDNA efficiently as shown in this work (Fig. 1b). We have shown that cNP may have unique scavenging properties due to its different extracellular and intracellular distribution characteristics from soluble polycation PDMA; it has higher cfDNA scavenge efficiency in the circulation and inflamed joint, and it shows a more effective reduction of DNA-mediated immune stimulatory activities in vitro and in vivo. This indicates that cationic nanostructured materials can modulate inflammation in an unexpected manner, which has not been reported before.

Although cNP might inhibit all NA-mediated activation of TLR, we have only focused on cfDNA in this study, because its high concentration in RA patients in both the serum and the synovial fluid has been documented[8–10]. In contrast, the concentration of dsRNA in the serum and the synovial fluid of RA patients is much lower than the cfDNA concentration[36], prompting us to overlook this parameter that may warrant future scrutiny. Using a known CpG that binds TLR9, we demonstrated that the cNP was more effective in blocking the TLR9 activation by the colocalization study (Fig. 1c and Fig. 2a). We then confirmed that cfDNA isolated from the synovial fluid of RA patients could stimulate human primary SFMC and FLS to up-regulate the synthesis of proinflammatory cytokines TNF-α and IL-6, respectively (Fig. 2d, e). Interference with cNP would inhibit the stimulation, with an efficiency higher than that of the soluble polycationic counterpart. This may be attributed to the cooperative effect of the cationic corona in binding the anionic cfDNA (Fig. 1b). This is consistent with our understanding of the intracellular trafficking of polyplexes and lipoplexes in nonviral gene delivery, that after internalization they would be sequestered into the endolysosomal compartments, where TLR9 resides in the vesicle membrane.

Using the CpG-induced acute RA model in mice, we confirmed that the scavenging approach using cNP would also work in vivo as supported by clinical scoring, micro-CT, MRI, and histology (Fig. 3). We then elaborated the merits of the concept with a clinically more relevant CIA model in rats. In addition to investigating the earlier stage therapy where the treatment was initiated at the onset of RA symptoms, we also evaluated the efficacy in the model at an established stage where treatment started one week after the onset and when the swelling in the hind paws peaked. In both cases, the treatment produced a positive effect (Fig. 4 and Fig. 5). This culminated in a rotational cage assay where the cNP-treated animals could recover partial mobility.

In all rat experiments, the cNP consistently out-performed the soluble counterpart. This may be attributed to a more favorable biodistribution of the cNP (Fig. 6a and Supplementary Fig. 9) due to the leaky microvasculature in the inflammatory sites[21,22]. In normal rats, free PDMA and cNP showed little difference in their accumulation in the joints and paws; the difference is in liver and the spleen as one would expect. In RA rats, both cationic materials have a higher accumulation in the joints and paws relative to normal rats, suggesting that inflammation does affect their biodistribution. Comparing the biodistribution of both materials in normal versus RA rats, there was also higher accumulation in the liver and kidney of the RA rats, suggesting those are also tissues with either inflammation or higher cfDNA level. Comparing the biodistribution of cNP versus PDMA in RA rats, more cNP had gone to the joints and paws, with a corresponding decrease in the lung, liver, and kidney. But there was an increase of cNP accumulation in the liver at 24 h versus 4 h as the cNP were cleared from the paws. Therefore, the biodistribution data elucidate why the cNP is a better performer in the RA model. However, recent findings indicate that the leaky vasculature observed in the tumor tissue of rodent is often irreproducible in human[37]. One must therefore be cautious in interpreting the importance of leaky vasculature in treating RA until this phenomenon is also confirmed in human.

The biodistribution on mechanistic speculation of how cNP is a more effective inflammation inhibitor than PDMA in vivo is nevertheless corroborated with the biochemical data. The cNP were more effective in scavenging the cfDNA in both the systemic circulation and the inflamed joints (Fig. 6b, c), consistent with the in vitro-binding data. The almost complete reduction of systemic cfDNA to the normal level by cNP is probably aided by the longer retention of the cNP in various tissues directly exposed to the systemic circulation. It is understandable that the reduction of cfDNA in the inflamed joints would not be as pronounced because of transport barriers to the cNP. Compared with PDMA, the cNP was also more effective in reducing the TNF-α expression in the inflamed joints (Fig. 6d). Taken together, the mechanistic studies validated the superior performance of the cNP in treating RA.

Toxicity of cationic materials is a concern in systemic applications. At a dose of 25 mg/kg, cNP produced the greatest relief of RA symptoms and did not induce thrombosis, whereas PDMA showed serious acute toxicity and caused blood coagulation and edema fluid in the lung (Supplementary Fig. 11a, b). Therefore, cNP accumulated less in lung and did not cause pulmonary edema. To further analyze the long-term toxicity of the cationic materials, we performed histological analysis of the heart, liver, spleen, lung and kidney of CIA rats after i.v. injection of cationic materials for 15 days on a daily basis. While PDMA caused different degrees of lesions in liver, spleen, lung, and kidney, the cNP-treated group showed no damage on various major organs by gross observation (Supplementary Fig. 12a). Also, the hepatotoxicity of cNP as reflected in ALP, ALT, AST levels and its

nephrotoxicity reflected in creatinine, urea, and uric acid before and after treatment were close to the normal group at the lower dose of 12.5 mg/kg (Supplementary Fig. 12b). The relatively low toxicity of cNP may be attributed to a biodistribution profile different from that of the soluble PDMA and an efficient NA scavenging capacity due to a multi-valent cationic corona. The latter allows the cNP to be administered at a lower dose in terms of charge density per mass compared with the soluble PDMA.

Motivated by the findings that cfDNA from RA patients would activate primary SFMC and FLS, we show that cNP can be used to scavenge pro-inflammatory NAs to treat experimental RA. While the balance between toxicity and efficacy remains to be optimized to define the translational potential of this therapeutic strategy for RA, this work suggests a new direction for nanomedicine. It is the opposite of using cationic materials to deliver NAs. All the innovations in nonviral gene delivery can be exploited for cNP designs to achieve efficacious and safe removal of pathogenic DAMP molecules. This new strategy may be widely applicable to tackle inflammatory diseases initiated by inappropriate activation of the TLR pathways.

## Methods

**Chemical regents.** Dodecanol terminated PLGA (composition 50:50, $M_n = 8000$, $M_n = 12,000$ calculated by [1]H NMR spectrum, Daigangbio Company, China) was dried by azeotropic distillation in the presence of toluene. 2-(Dimethylamino)ethyl methacrylate (DMA, purity > 98.5%, TCI) was dried over $CaH_2$ and distilled before use. Bromoisobutyryl bromide (purity 97%, Alfa), N,N,N',N'',N''-penta-methyldiethylenetriamine (PMDETA, purity > 98%, Acros), cuprous bromide (CuBr, purity 99%, Aladin, China), 2-aminoethyl methacrylate hydrochloride (AEMA, Sigma Aldrich), triethylamine (Guangzhou Chemical Reagent, China), bis (tert-butyl)dicarbonate (Adamas), trifluoroacetic acid (TFA, purity > 99%, Aladdin, China), and ethyl α-bromoisobutyrate (EBiB, purity > 98%, TCI) were commercial products and used directly without further treatment. Fluorescein isothiocyanate (FITC, Aladdin, China) was used in its stock solution, 10 mg/mL, in DMSO. Alexa Fluor® 750 NHS Ester (succinimidyl ester) (AF750-NHS, Invitrogen) was dissolved in DMSO to prepare a stock solution of 10 mg/mL.

Boc-protected AEMA monomer (AEMA-Boc) was used for copolymerization with DMA in order to introduce fluorescence probe. It was synthesized according to literature[38]. The purity was proved by [1]H NMR (Bruker AVANCE III 400 MHz, in CDCl3) spectrum δ (ppm): 6.1 and 5.6 (s, 2H, CH2 = C), 4.8 (br, 1H, NH–C = O), 4.2 (t, 2H, NCH2CH2O), 3.4 (br, 2 H, NCH2CH2O), 1.9 (s, 3H, CH3-C), 1.5 (s, 9H, (CH3)3-C).

**Synthesis of PDMA470.** PDMA470 was prepared by atom transfer radical polymerization (ATRP) (see schematic syntheses in Supplementary Figure 1b). 14.3 mg (0.073 mmol) EBiB, 10.8 mg (0.073 mmol) CuBr, 13.0 mg (0.073 mmol) PMDETA, 6.35 g (40.4 mmol) DMA, 84 mg (0.36 mmol) AEMA-Boc, and 2.5 mL of anisole were charged into a dried polymerization tube and the mixture was deoxygenated by three freeze-pump-thaw degassing cycles. The polymerization was conducted in an oil bath at 60 °C for 17 h and was terminated by cooling and exposing to the air. The resulting mixture was diluted with 40 mL of THF and the solution was then passed through a short alkaline alumina column to remove the copper residue. After evaporated to dryness, the crude product was dissolved in 20 mL of THF and precipitated dropwise into 200 mL of cool hexane and repeated three times. The collected product was dried in vacuum for 24 h at room temperature. The monomer conversion rate was 85% as determined by composition of the monomer and PDMA in the polymerization mixture from [1]H NMR (Bruker AVANCE III 400 MHz) spectrum in CDCl3. Thus, the degree of polymerization was obtained as 470 and the molecular weight ($M_n$) of PDMA was calculated to be 71000 Dalton. The [1]H NMR spectrum of the pure product was shown in Supplementary Figure 1d. The polydispersity, $M_w/M_n = 1.32$, was obtained by gel permeation chromatography (GPC, Agilent) using DMF as the eluent (Supplementary Figure 1e). To remove the Boc group, a suspension of the Boc product (0.056 mmol) in 10 mL of anhydrous CH2Cl2 was stirred for 10 min in an ice-water bath and 10 mL of TFA was added. After the reaction was carried out overnight, TFA was removed under reduced pressure. The crude product was dissolved in 10 mL of water and the solution was dropped into 200 mL of cool diethyl ether to precipitate product. The product was redissolved in 20 mL of water and precipitated dropwise into 500 mL of NaOH solution (pH = 14). The product was isolated by filtration and then dissolved in 30 mL of 1 M HCl. The solution was titrated to pH 7 with a dilute solution of NaOH, and then dialyzed against deionized water for 24 h. The collected solution was lyophilized (Alpha 1–4 LD, Christ) for 24 h.

**Synthesis of PLGA-b-PDMA463 diblock copolymer.** PLGA-b-PDMA463 was prepared by ATRP in two steps (see schematic syntheses in Supplementary

Figure 1c). The first step was to synthesize PLGA-Br macroinitiator. 3 g (0.25 mmol) of PLGA was dissolved in 18 mL of CH2Cl2, and then 60 μL (0.4 mmol) of triethylamine was charged. To this solution being cooled to 0 °C, 50 μL (0.4 mmol) of bromoisobutyryl bromide in 2 mL of CH2Cl2 was added dropwise. The solution was stirred in an ice bath for 4 h and then was brought to 30 °C for 20 h. The reaction mixture was washed with aqueous solutions of 1 M HCl, saturated NaHCO3, and saturated NaCl. The organic phase was dried with magnesium sulfate, concentrated to 10 mL, and then precipitated twice into 100 mL mixture of cold diethyl ether and methanol (volume ratio = 2:1). The product was isolated by filtration and dried in vacuum for 24 h at room temperature.

The second step was the ATRP block copolymerization. 0.55 g (0.046 mmol) PLGA-Br, 6.6 mg (0.046 mmol) CuBr, 8.2 mg (0.046 mmol) PMDETA, 3.71 g (23.6 mmol) DMA, 53.4 mg AEMA-Boc (0.23 mmol), and 3 mL of toluene were charged into a dried polymerization tube. The polymerization was carried out at 70 °C for 52 h. The rest procedure was the same as that of PDMA preparation. The monomer conversion was 89% as estimated by [1]H NMR spectrum in CDCl3 and thus the degree of polymerization of the PDMA block was determined as 463 (Supplementary Figure 1d). Therefore, the PLGA-b-PDMA463 sample was given. The $M_w/M_n$ was 1.21 as given by GPC using DMF as the eluent (Supplementary Figure 1e). The Boc group was removed similarly as the previous described.

**Fluorescent labeling of cationic materials.** FITC or AF750-NHS was applied to label PDMA and PLGA-b-PDMA, respectively (see schematic syntheses in Supplementary Figure 1b and c). A general procedure was given. FITC and PDMA (mass ratio 1:10) were dissolved in DMSO and a solution of PDMA, 1 mg/mL, was obtained and stirred for overnight at room temperature. The solution was added with deionized water and concentrated to 2.5 mg/mL using ultrafiltration tube (Amicon® Ultra-4 10K, Millipore).

**Preparation of cNP by self-assembly.** PLGA-b-PDMA463 diblock copolymer was dissolved in THF at a concentration of 12.5 mg/mL. Under ultrasound (Ultrasonic Processor, Sonics VCX105), the solution of block copolymer was dropped slowly to an aqueous solution of hydrochloric acid at pH = 3 until the polymer concentration reached 2.5 mg/mL. The solution was dialyzed against a diluted hydrochloric acid at pH=3 for 12 h to remove the THF. Then, the pH was adjusted to 7.4 by adding a mixture of Na2HPO4·12H2O and KH2PO4 and the phosphate concentration of the final PBS buffer was 10 mM.

**Characterization of cNP.** The size and polydispersity index (PDI) of cNP in 10 mM PBS buffer (pH = 7.4) were measured in triplicated by dynamic light scattering (DLS, Malvern Zetasizer Nano ZS) at 25 °C. The number average diameter of cNP was 44.0 nm with PDI being 0.149 (Supplementary Figure 1f). The zeta potential of cNP in PBS (pH = 7.4) was 18.5 ± 3.7 mV measured by Malvern Zetasizer Nano ZS. Furthermore, the morphology of cNP was characterized by transmission electron microscope (TEM, JEOL1400+) (Supplementary Figure 1g).

**Bio-reagents.** Phosphate buffered solution (PBS, pH = 7.4), Dulbecco's modified Eagle medium (DMEM), Iscove's modified Dulbecco's media (IMDM), Roswell Park Memorial Institute 1640 Medium (RPMI 1640), and fetal bovine serum (FBS) were purchased from United States Origin, Gibco.

Calf thymus DNA was purchased from Sigma Aldrich and its stock solution, 1 mg/mL, was prepared in PBS. CpG 2006, CpG 1668, CpG 1826, and Cy5-CpG 1826 were purchased from Genscript China. The stock solutions of un-labeled CpG in PBS were prepared as 100 μg/mL, while that of the labeled CpG was 70 μg/mL. cfDNA from patients was extracted from the synovial fluid of RA patients in General Hospital of Guangzhou Military Command of PLA, according to a protocol described below. The studies were approved by the Ethics Committee of the School of Life Science, Sun Yat-sen University.

Mouse TNF-α ELISA Kit (Biolegend), Human TNF-α ELISA Ready-SET-Go! (eBioscience), and Human IL-6 ELISA Ready-SET-Go! (eBioscience) were purchased accordingly.

Anti-MyD88 antibody (MyD88, cat. # ab2064), anti-TRAF6 antibody (TRAF6, cat. # ab40675), anti-β-actin antibody (β-actin, cat. # ab8227), anti-TNF alpha antibody (cat. # ab6671), anti-IL 6 antibody (cat. # ab9324), anti-MMP-3 antibody (cat. # ab53015), goat anti-mouse IgG H&L (HRP) (cat. # ab6789), and goat anti-rabbit IgG H&L (HRP) (cat. # ab6702) were purchased from Abcam.

Primers of glyceradehyde-3-phosphate dehydrogenase (GAPDH) were purchased from Sangon Biotech, China. Other primers were synthesized by Sangon Biotech, China.

Pam3CSK4, polyinosinic-polycytidylic acids (poly (I:C)) (HMW), ssRNA40, R848 and QUANTI-Blue™ were purchased from Invivogen. Hyaluronidase from bovine testes (Sigma), ficoll-paque plus (GE Healthcare), Lipofectamine® 2000 (Lipo, Invitrogen), Lysotracker Red DNA-99 (Yeasen, China), DAPI (Invitrogen), Quant-iT™ PicoGreen™ dsDNA Assay Kit (PicoGreen, Invitrogen), Radio-Immunoprecipitation Assay Lysis Buffer (RIPA Lysis Buffer, Beyotime, China), polyvinylidene fluoride (PVDF) membrane (pore size 0.45 μm, 26.5 cm × 3.75 m roll, Millipore), Electrochemiluminescence Detection Kit (ELC, Beyotime, China), Tissue-Tek Optimal Cutting Temperature Compound (O.C.T. Compound, Sakura), Diaminobenzidine (DAB) Plus kit (Maxin Biotechnologies), Dynabeads®

SILANE Viral NA Kit (Invitrogen), Circulating Cell-free DNA Purification Kit (Qiagen), TRIzol™ reagent (Invitrogen), Rneasy-universal tissue kit (Qiagen), PrimeScript RT reagent Kit With gDNA Eraser and oligo (dT) (Takara) and SYBR® Premix Ex Taq™ II kit (Takara), Immunization grade bovine type II collagen solution (2 mg/mL, Chondrex), complete Freund's Adjust (5 mg/mL, Chondrex), incomplete Freund's Adjust (5 mg/mL, Chondrex), chloral hydrate (Energy, China), and isoflurane (RWD Life Science, China) were purchased from companies.

The stock solution of 3-(4,5-Dimethyl-2-thiazolyl)-2,5-diphenyl-2-H-tetrazolium bromide (MTT, purity 98%, Sigma Aldrich) in PBS was prepared as 5 mg/mL. A stock solution of EtBr (Sigma Aldrich) in PBS was prepared as 1 mg/mL. Haematoxylin solution was prepared by dissolving 4 g of hematoxylin (Aladdin, China), 200 g of aluminum potassium sulfate dodecahydrate (Aladdin, China), and 0.8 g of sodium iodate (Aladdin, China) in 1280 mL of 6.25% ethanol. Eosin solution was prepared by dissolving 2.5 g of Eosin Y (Aladdin, China) in 1000 mL of 95% ethanol with 0.5 mL acetic acid.

**Cell**. Raw 264.7 cells were purchased from ATCC and cultured in DMEM with 10% heat-inactivated FBS and antibiotics at 37 °C in a humidified atmosphere with 5% $CO_2$. Ramos Blue™ reporter cells were purchased from Invivogen, USA, and maintained in IMDM with 10% heat-inactivated FBS and antibiotics at 37 °C, 5% $CO_2$ in a humidified chamber. These two cell lines have been authenticated via application in multiple labs without any problem. Primary synovial fluid mononuclear cells (SFMC) were enriched in the inflamed RA joint as described in literature[39]. Briefly, SFMC was collected as following steps. First, 1 unit of hyaluronidase was added into each mL of synovial fluid to reduce the viscosity. Then the debris were removed by filtration through 200 mesh screen cloth, and the mononuclear cells were enriched from synovial fluid using Ficoll-paque plus and cultured with completed RIPM 1640 medium in a humidified incubator at 37 °C under 5% $CO_2$. Primary RA fibroblast-like synoviocytes (FLS) were obtained under the approved clinical protocol of Prof. Hanshi Xu and grown in a DMEM/F12 medium containing 10% heat-inactivated FBS, in a humidified incubator at 37 °C under 5% $CO_2$[40]. RA-FLS were used from passage five to nine, and the cell type had been identified as described in their previously published work[40]. The primary SFMC and FLS extracted from RA were approved by the Ethics Committee of the School of Life Science, Sun Yat-sen University and General Hospital of Guangzhou Military Command of PLA. All the cells have been tested for mycoplasma contamination using PlasmoTest™—Mycoplasma Detection Kit (Invivogen).

**Animals**. Female Lewis rats (7 weeks) and BALB/c mice (6–8 weeks) were purchased from Beijing Vital River Laboratory Animal Technology. The sample size for the animal study was estimated after consultation with the biostatistics service at Columbia University. All rats and mice were bred, housed and used under specific pathogen-free conditions in the animal facility of the School of Life Science, Sun Yat-sen University. The animal studies were performed with the approval of the Ethics Committee of the School of Life Science, Sun Yat-sen University.

**Human subjects**. Studies were approved by the Ethics Committee of the School of Life Science, Sun Yat-sen University, and the Ethics Committee of General Hospital of Guangzhou Military Command of PLA. Besides, the informed consent was obtained from all related patients. The patients meet the criteria from American College of Rheumatology from physical examination center and their basic information was shown in Supplementary Table 1. The recruited patients gave written consent according to a protocol approved by the Ethics Review Committee of the General Hospital of Guangzhou Military Command of PLA.

**DNA extraction and measurement**. For extraction of cfDNA from RA patients, the synovial fluid from RA patients was collected in heparin sodium-containing tubes. The synovial fluid was centrifuged at 400×g for 15 min at 4 °C to remove the cells. The supernatant was re-centrifuged at 12,000×g for 10 min to remove the cell-debris and then incubated with bovine hyaluronidase (1 units each mL) for 30 min at 37 °C to reduce the viscosity. The obtained supernant was used to extract cfDNA with QIAamp Circulating Nucleic Acid Kit (Qiagen).

For extraction of cfDNA in plasma of rats, peripheral blood sample of rats of different groups were taken from the eye socket at different days and collected in EDTA-containing tubes. Then the samples were centrifuged at 400×g for 10 min at 4 °C and then the plasma fraction was re-centrifuged at 12,000×g for 10 min at 4 °C to remove cell debris, which was stored at −80 °C for analysis. The cfDNA was extracted from 100 μL of plasma using Dynabeads® SILANE Viral NA Kit. For cfDNA in inflamed joints, the rats were sacrificed at the day 29 and their synovial fluids were collected from the joints by continuous washing articular cavity with PBS. The combined PBS was centrifuged at 400×g for 15 min at 4 °C to remove the cells. The obtained supernatants were used to extract cfDNA with Circulating cfDNA Purification Kit. The concentration of cfDNA was determined with Quant-iT™ PicoGreen™ dsDNA Assay Kit.

**Binding efficiency of cationic materials and NAs**. The binding ability of cNP and PDMA with NA was evaluated according to the method in literature[41]. Briefly, the calf thymus DNA solution (4 μL, 1 mg/mL of stock solution in PBS) and the EtBr

solution (4 μL, 1 mg/mL of stock solution in PBS) were mixed firstly. To which, different volume of the stock solutions of the cationic materials and 16 μL of FBS or PBS were added. The final volume was adjusted to 160 μL by adding fresh PBS. After incubation at 37 °C for 24 h, 100 μL of supernatant containing the remaining DNA/EtBr complex of each dose was transferred to a 96-well plate. The fluorescence intensity of the complex at wavelength of 590 nm was measured by excitation at wavelength of 485 nm with the Multiwall Plate Reader (BioTek Synergy2 Gen 5). The NA-binding efficiency with cationic materials was evaluated by $(1-(A-A_0)/(A_1-A_0)) \times 100\%$, where $A$ was the fluorescence intensity of EtBr/DNA complex of supernatant after adding materials, $A_0$ was the EtBr fluorescence intensity, and $A_1$ was the fluorescence intensity of EtBr/DNA complex.

**MTT cytotoxicity assay**. Raw264.7 cells were cultured in DMEM with 10% heat-inactivated FBS and antibiotics at 37 °C in a humidified atmosphere with 5% $CO_2$. $10^4$ cells/well were plated in a 96-well plate. After 12 h, the medium was replaced with the medium containing cationic materials of different concentrations. After incubation for 24 h, MTT reagents were added to the wells and incubated for another 4 h. The supernatant was removed and DMSO was added to dissolve the residue. The absorbance of solution was recorded at 570 nm with the Multiwall Plate Reader.

**Inhibition of TLR by extracellular agonists**. First, to study the inhibition of TLR2, 3, 7 and 9 activation by cationic materials in Ramos Blue™ reporter cells, 1 μM of Pam3CSK4, Poly (I:C) (HMW), R848 and CpG ODN 2006 were incubated with the cationic materials at different concentrations, respectively, and then the mixture was added to a 96-well plate in which $5 \times 10^4$ Ramos Blue™ reporter cells/well were plated in advance. After incubation for 24 h, the supernatants were collected and the QUANTI-Blue assay was performed to evaluate TLR activation according to the protocol suggested by the manual instruction. In brief, after the supernatant was incubated with the QUANTI-Blue™ medium for 1 h, the medium changed to purple–blue color due to embryonic alkaline phosphatase (SEAP) activity, which was quantified by the OD at 650 nm with a Multiwall Plate Reader. The TLR activation was evaluated by $(X-X_0)/(X_1-X_0) \times 100\%$, where $X$ was the OD value of the experimental group, $X_0$ the control group, and $X_1$ the free TLR agonist group.

Second, to study the cationic material inhibition of immune stimulatory effects caused by different agonists in RAW264.7 cells, 1 μM of Pam3CSK4, 1 μM of Poly (I:C) (HMW), 1 μM of R848, 2 μM of ssRNA40, and 1 μM of CpG 1826 with 25 μg/mL cationic materials were added into a 96-well plate with $2 \times 10^4$ RAW264.7 cells/well.

To determine whether the cationic materials might bind to the TLR7 receptor, first RAW264.7 cells were treated with 2 μM of ssRNA40 and 25 μg/mL cationic materials, and then 1 μM of R848 was added one hour later. The supernatants were collected after 24 h incubation and the TNF-α expression level was measured using ELISA kit.

**Inhibition of TLR9 by intracellular agonists**. To test if the cationic materials could competitive bind the NA from the TLR9, CpG 2006 was incubated with the Ramos Blue™ reporter cell in the 96-well plate for 4 h in advance, then the materials were added. After incubation for 24 h, the supernatants were collected and the QUANTI-Blue assay was performed as previously described.

**Western-blotting**. After the Ramos Blue™ cells were incubated with CpG 2006 in the absence or presence of cationic materials, cell lysates in RIPA Lysis buffer were boiled and centrifuged, and the supernatant was then separated by 10% SDS PAGE. The proteins on the PAGE were transferred to a PVDF membrane and probed with the antibodies against MyD88 (1:1000), TRAF6 (1:2000), and β-actin (1:2000), followed by the HRP-conjugated Goat Anti-Rabbit IgG second antibody (1:5000). The hybridized bands were then visualized with ELC assay by FluorChem R (ProteinSimple, America). Quantitative analysis of the western-blot was done by using Software AlphaView SA.

**Blocking internalized DNA stimulation cells**. $10^4$ RAW264.7 cells in 100 μL complete medium/well were plated in 96-well and then 1 μM CpG 1826 was added to incubate for 4 h. After washing three times with PBS to remove the excessive CpG in the medium, the cationic polymers (25 μg/mL) were added. After incubation for 24 h, the supernatants were collected and TNF-α expression level was determined. For evaluating the cationic materials inhibiting the stimulation of internalized NA, the cfDNA from RA patients was delivered by Lipofectamine 2000. SFMC derived from RA patient ($5 \times 10^4$ SFMC/well) was first transfected with 1 μg of cfDNA from the same RA patient using Lipofectamine 2000 in 96-well plate. After 4 h, the medium was replaced with fresh medium containing cationic materials of 25 μg/mL. After incubation for 24 h, the culture supernatant was collected to evaluate the TNF-α expression level, which was mainly from the synovial macrophages[42,43]. For stimulating primary FLS with cfDNA, the same procedure was performed, whereas $5 \times 10^3$ primary FLS/well were plated in 96-well plate; the optimized dose, 0.5 μg/mL, of cationic materials was added. Then IL-6 expression level was measured, which was the major product of FLS[42,44]

**Cytokine concentration analysis with ELISA**. The concentration of TNF-α in the culture supernatants of RAW cells was determined with ELISA Kit using Mouse TNF-α ELISA Kits. Also, the concentrations of TNF-α and IL-6 in the culture supernatants of primary cells from patients were determined with ELISA Kits using human TNF-α and IL-6 ELISA Kits.

**Cellular colocalization of CpG and cationic materials**. In the NUNC™ Lab-Tek™ eight-well plates, $2 \times 10^4$ cells/well of RAW264.7 cells were cultured overnight. Then two experiments were conducted. The first one was to test if the cationic materials could reduce the cell uptake of extracellular NA. 1 μM of Cy5-CpG 1826 with 1.0 μg/mL cationic materials were added into the new culture media to replace the old media and incubated for 12 h. The second one was to test if the cationic materials could bind intracellular NA agonist and inhibit its stimulation to TLR9. After washing three times with fresh culture medium, 1 μM of Cy5-CpG 1826 was added into the culture media and incubated for 4 h. The excessive CpG was removed by washing three times with PBS, then the cationic materials labeled with FITC in medium (1.0 μg/mL) were added. After 4, 8, 12 h, the cells treated in these two experiments were stained with LysoTracker Red DNA-99 and DAPI for confocal microscopic observation (Leica SP8). Mean fluorescence intensity of Cy5-CpG 1826 was the optical density of Cy5-CpG 1826 per cell, as calculated by Image J software. Colocalization ratio of Cy5-CpG 1826 and FITC-cationic materials was the ratio of colocalization area (the area of white spots) and foreground area (the area of green and purple spots), calculated by Leica Application Suite.

**Animal model induction and treatment**. The acute arthritis model was induced with CpG 1668[26,45]. Six micrograms of CpG was injected intra-articularly into the right knee joints of BALB/c mice. Onset of arthritis occurred one day later, and the arthritic mice were evenly divided into model and treatment groups randomly. There were three groups including normal control, model control and cNP treated group, with five mice per group. For cNP treated group, a solution of cNP in PBS (2.5 mg/mL, 100 μL), with dose 12.5 mg/kg, was given daily by intravenous injection for 7 days.

The CIA model was established according to literature[46]. Female Lewis rats were given intradermal injections of bovine type II collagen with the Freund's adjuvant (emulsion of two reagents at volume ratio = 1:1), one site at the tail base (0.1 mL) and two sites at the back (0.2 mL for each site) on day 0 and day 7. The onset of arthritis occurred on day 13, and the arthritic rats were evenly divided into model and treatment groups at random. The effect of cationic polymers was evaluated in both early stage and established stage of RA. For the early stage RA progression, the treatment began at day 13 with five groups animal. The rats without disease were applied as the normal control ($n = 5$). The CIA rats without treatment were the model control ($n = 10$). The other three treatment groups, 10 rats/group, were the treatment by PDMA (12.5 mg/kg) and cNP (12.5 and 25 mg/kg). For the established stage treatment, the normal and model groups were the same as above, whereas for the treatment group, we selected a dose of 25 mg/kg cNP ($n = 5$) to evaluate its efficacy.

**Joint swelling measurements and clinical scores**. The swelling and clinical scores of joints were evaluated daily from the onset of arthritis (day 1 for mice and day 13 for rats), until the animals were sacrificed. The investigator was blinded to the group allocation when measuring the joint swelling. For the acute RA model, the swelling of the joints in mice was evaluated by measuring the diameter of the right ankle with a vernier caliper (Guanglu, China). For the CIA model, the swelling of the joint of the rats was evaluated by measuring the volume of hindpaws with the Plethysmometer (YSL-7C, Yiyan Tech, China) and calculating the average volume. Animals died during the experiments were excluded from the analysis.

Clinical scores of each hindpaws and forepaws of rats were obtained following the standard evaluation process[27]. Score 0: no evidence of erythema and swelling occurred. Score 1: erythema and mild swelling appeared. Score 2: erythema and mild swelling extended from the ankle to the tarsals. Score 3: erythema and moderate swelling extended from the ankle to metatarsal joints. Score 4: erythema and severe swelling encompassed the ankle, paws, and digits or ankylosis of the limb. Then the average clinical scores of hindpaws and the average clinical scores of forepaws were calculated.

**Photograph**. For evaluation of joint swelling, the hindpaws of mice were photographed by camera at day 3 post articular injection of CpG. The hindpaws of rats were photographed at day 13 (before treatment), day 16 (after treatment for 3 days), and day 19 (after treatment for 6 days) by camera.

**Micro-CT imaging**. To evaluate the bone damage of ankle joints, the ankle joints of mice sacrificed at day 8 and rats sacrificed at day 29 were fixed in 10% buffered formalin for a week and scanned at 80 kV and 500 μA with the resolution of 19 μm in ex vivo micro-computed tomography (Micro-CT, Siemens Inveon Micro-CT/PET) for 50 min. Then the dataset was reconstructed using Inveon Research Workplace to obtain the 3D images of joints and to measure BMD and other morphometric parameters. The calcaneus was evaluated for bone destruction and its BMD was measured as a comparative indicator of bone damage. The region of

interest (ROI) in calcaneus as highlighted in Supplementary Figures 7, 8 was chosen for analysis with the following morphometric parameters: (1) bone volume fraction: trabecular bone volume/total volume. Total volume is the volume within which the measurement is performed. (2) Ratio of the segmented bone surface to the segmented bone volume: bone surface area/bone volume. (3) Mean thickness of trabeculae. (4) Mean distance between trabeculae: trabecular spacing. (5) Ratio of convex to concave surface: trabecular pattern factor. A higher concavity represents a well-connected spongy lattice, whereas a higher convexity indicates a poorly connected trabecular lattice in two-dimensional sections.

**MRI imaging**. Magnetic resonance imaging (MRI, Achieva 3.0T, Philips Healthcare, Netherlands) was performed to characterize the development of joint effusion, soft tissue swelling and lymph node volume after treatment at day 7 for mice and day 24 for rats. Different groups of animals were anesthetized by i.p. injection of chloral hydrate (3.5% for mouse, 10% for rat). After 10 min, their hindpaws and knee joints were visualized by performing MRI utilizing an eight-channel transversal animal coil. Coronal fat-suppressed T2W TSE was acquired with the following parameters: time of repeat/time of echo, 800/80 ms; field of view, $55 \times 55$ mm²; slice thickness, 1 mm; slice number, 8; number of signal average, 1; matrix, $176 \times 190$. Spectral pre-saturation inversion recovery (SPIR) fat suppression technique was used. The total imaging time of this sequence was about 4 min 19 s.

**Histological analysis and immunohistochemical staining**. After the animals were sacrificed at day 8 for mice and at day 29 for rats, the ankle, knee, digital and wrist joints of scarified animals as well as their heart, liver, spleen, lung and kidney were fixed in 10% buffered formalin and then joints were incubated in decalcifying solution (4% hydrochloric acid in 4% formaldehyde) at room temperature for 7 days for decalcification. After paraffinization, microtome (Leica) slices of 2 μm were prepared for histological examination. The slices were stained with haematoxylin and eosin, respectively, and the inflammatory cell accumulation in synovial tissues, bone and cartilage was evaluated by Vectra Automated Quantitative Pathology Imaging System (PerkinElmer). For immunohistochemical staining, after deparaffinization the slices were subjected to antigen recovery in 0.01 M sodium citrate buffer at 125 °C for 30 s, followed by 10 s at 90 °C, and then subjected to the endogenous peroxidase inactivation by covering tissue with 3% hydrogen peroxide for 5 min. After blocking non-specific binding sites with 10% goat serum in PBS, the slices were incubated with different biotinylated monoclonal antibody or polyclonal antibody with a dilution rate of 1:100, respectively, at 4 °C for 24 h. Then the slices were incubated with horseradish peroxidase (HRP)-conjugated secondary antibody with a dilution rate of 1:800 at 37 °C for 1 h. Sections were developed using the DAB substrate and then counterstained with haematoxylin. The biotinylated monoclonal antibody or polyclonal antibody used were anti-TNF alpha antibody, anti-IL 6 antibody, and anti-MMP-3 antibody. The images were captured and analyzed by Vectra Automated Quantitative Pathology Imaging System. The positive results (brown staining) were evaluated using Nuance 3.0.2 and inform 2.1.1 software.

For evaluation of acute toxicity of cationic materials, the lungs of normal rats and treated rats after i.v. injection of 25 mg/kg of cNP were photographed at 24 h immediately after the rat sacrifice. And the lung of treated rat after i.v. injection of 25 mg/kg of PDMA was photographed immediately after it died within 10 min. The lungs of sacrificed rats were then embedded in O.C.T. compound, quickly frozen at −80 °C and then sectioned at 6 μm. H&E histological examination of hyperemia was evaluated by the Vectra Automated Quantitative Pathology Imaging System.

For evaluation of histology scores of the joints of mice or rats, each histopathologic feature was graded by a trained pathologist (SML) using a scoring system as previously described[47]: synovial cell lining hyperplasia (0–2); pannus formation (0–3); mononuclear cell infiltration (0–3); polymorphonuclear leukocyte infiltration in periarticular soft tissue (0–3); cellular infiltration and bone erosion at distal tibia (0–3); and cellular infiltration of cartilage (0–2). The histology score of each animal joint was the sum of all the histopathologic feature scores.

**Rotational cage test study**. At the end of treatment (day 29), the move ability of normal, model and cNP established stage treated rats ($n = 3$ per group) were placed on a programmable, motorized wheel apparatus (21 cm diameter, 40 cm length, made in South China University of Technology, Guangzhou, China), and then the cage rotated based on the protocol according previous report[48]. The normal and model group were used as control. All rats in this study were given pre-conditioning exercise before the rotation cage test. The process of their running against the rotation of the cage was recorded as a video. Moreover, the maximum speed of rotation at which the rats could manage was also recorded.

**Examination of liver and renal function of rats**. Peripheral blood sample of rats of different groups were taken from the eye socket at day 13, 29, and 60 after first collagen II injection. After 2 h, the samples were centrifuged at 800×g for 10 min. The plasma fraction was re-centrifuged at 800×g for 10 min again to obtain cell-free serum and stored at −80 °C for analysis. The levels of ALP, ALT, AST, creatinine, urea, and uric acid were measured by KingMed Diagnostic Company.

**Real-time PCR**. At day 29, the shinbones of sacrificed rats were removed and homogenized in TRIzol™ reagent with a homogenizer (T 18 digital ULTRA-TURRAX, IKA) according to the reference[17]. Briefly, total RNA was extracted using Rneasy-universal Tissue Kit, then 2 µg of RNA was reverse-transcribed into cDNA in a total volume of 20 µL using PrimeScript RT reagent Kit With gDNA Eraser and oligo (dT). The cDNA levels were measured by SYBR green real-time in the Lightcycler (Applied Biosystems QPCR) and were normalized with that of glyceraldehyde-3-phosphate dehydrogenase (GAPDH). The relative mRNA levels of cytokines and MMP-3 in normal group were set as 1. The sequences of the primers were as follow:

qTNF-α FP: 5′-TGATCCGAGATGTGGAACTG-3′;
qTNF-α AP: 5′-CGAGCAGGAATGAGAAGAGG-3′;
qIFN-α FP: 5′-GTCTTCACACTCCTGGCACA-3′;
qIFN-α AP: 5′- GCTTGAGCCTTCTGGATCTG-3′;
qIL-6 FP: 5′-TGCCTATTGAAAATCTGCTCTGG-3′;
qIL-6 AP: 5′- CATTGGAAGTTGGGGTAGGAAG-3′;
qMMP-3 FP: 5′- CTGGAATGGTCTTGGCTCAT-3′;
qMMP-3 AP: 5′-GAGCAGCAACCAGGAATAGG-3′.

**Biodistribution of cationic materials**. At day 30 after the first immunization, the untreated CIA rats and normal rats were anaesthetized (1.5–2% isoflurane, 0.5 ml/min oxygen) and the Alexa Fluor® 750 labeled cationic materials were intravenously injected. The biodistribution of the materials in four groups, including PDMA-treated normal group, PDMA-treated model group, cNP-treated normal group, and cNP-treated model group, was monitored by NIRF imaging using an in vivo imaging scanner (Carestream FX PRO) at different time points during 24 h. At 4 and 24 h, one from each group was sacrificed for dissection. The joints, thymus, heart, lung, liver, kidney, pancreas, spleen, and bladder were taken out for ex vivo NIRF imaging and their mean NIRF intensity was calculated by Bruker MI software.

**Statistical analysis**. Statistical significance of survival curve of rats was calculated by log-rank (Mantel–Cox) test using GraphPad Prism 6.0, and statistical analysis of other experimental data was performed by one-way ANOVA with LSD post-test using IBM SPSS Statistic 22.

## Data availability

All data supporting the findings of the current study are available in the article and its Supplementary Information. Additional relevant data are available from the corresponding author upon reasonable request.

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

## Acknowledgements

Financial support from the Guangdong Innovative and Entrepreneurial Research Team Program (2013S086), the National Natural Science Foundation of China (21875290, 51403243, 51533009 and 51820105004), Natural Science Foundation of Guangdong Province (2014A030312018) is gratefully acknowledged. Partial support to K.W.L. by NIH AI1096305 is also acknowledged.

## Author contributions

H.L. conducted the main experiments; B.P. assisted with DNA binding and cell inhibition assays; C.D. assisted with biological experiments; L.L. designed and supervised all the experiments and wrote manuscript; J.M. conducted medical imaging; J.S. supplied imaging and analysis; S.W. and X.W. supplied RA samples; H.X. and X.G. provided discussion on RA; K.W.L. and H-Q.M. supervised and wrote manuscript; Y.C. designed materials, supervised and wrote manuscript.

## Additional information

**Competing interests:** The authors declare no competing interests.

