## [Peer Review File · Nature Communications]

Reviewers' comments:

Reviewer #1 (Remarks to the Author):

Liang et al presented a novel approach to use cationic nanoparticles for treatment of arthritis. A simple cationic PLGA-PDMA polymeric nanoparticle was shown to treat arthritis more effectively than a soluble PDMA polymer. Series of in vitro studies were conducted to fully confirm the cell-free DNA (cfDNA) adsorption mediated therapeutic response while three different in vivo models on mice and rats demonstrated significant efficacy against arthritis with detailed histology analysis. The simplicity of the utilized material and the excellent treatment outcome clearly demonstrated the significance of the work. I would recommend the acceptance of the paper if the authors can address the following minor points.

1. The binding of the cNP with cfDNA is mainly through charge interaction. It is well known that highly charged nanoparticles may adsorb serum proteins upon injection resulting in neutralization of the surface charge. It is recommended that the authors should demonstrate the DNA binding of the cNP in the presence of serum or serum albumin to confirm that the protein coating would minimally impact the cNP-cfDNA interaction as well as other biological functions of the cNP.
2. Quantitative results for the biodistributions of all major organs are desired based on the images shown in Figure 6a.
3. Histology analysis all major organs other than lung (in Supplementary Figure 8.) should be provided to further confirm the toxicity profile
4. The chemical structure of PLGA in supplementary document should be more accurate. It is a random copolymer so an "r" should be written between the repeating units.
5. Figure S6: Although it is not the focus of the figure, it should be acknowledged that the NIR fluorescence from major organs may not be observed due to the presence of the fur.

Reviewer #2 (Remarks to the Author):

This study shows that ~40 nm cationic nanoparticles can scavenge cfDNA derived from RA patients to inhibit the activation of primary synovial fluid monocytes (SFMC) and fibroblast-like synoviocytes (FLS). They also show that intravenously injection of cNP into a CpG-induced mouse model or collagen-induced arthritis (CIA) rat model can relieve RA symptoms with respect to ankle and tissue swelling as well as bone and cartilage damage. The in vivo results are very well characterized. methods very detailed and the concept very promising as a new therapeutic approach in RA. However the mechanistic experiments to explain specificity of the cNP and neutralization of cfDNA are not convincing.

- Can you please mention in the introduction the link between cell-free DNA and NETs, that has also been linked to rheumatoid arthritis?.
- Please elaborate the rationale to use cNP instead of other options studied in previous studies (Lee et al, 2011). If the main novelty of this manuscript is the improved material used in this study, studies that compare both materials are needed.
- Also for the readers not familiar with NP, could you please elaborate why these materials bind DNA? Can they bind to something else? Meaning can they work as scavengers of other negatively charged pro-inflammatory molecules released in the extracellular space from dying cells?
- More complete studies using different TLRs reporters are needed to show that the new material is only inhibiting nucleic acid sensing TLRs (TLR3, TLR7 and TLR9 vs TLR2 or TLR4 or other stimuli like TNF/IL1). Also experiments and Raw and FLS using different ligands including mammalian DNA and the cNP are needed to suggest TLR specificity.
- The authors stated that cNP inhibits intracellular agonists to TLR9. Further experiments using for instance chloroquine, that is an inhibitor of endosomal acidification, are needed to prove that cNP do not neutralize extracellular inflammatory nucleic acids. Is this the mechanism, when you add the cNP and CpG at the same time? Is the cNP also decreasing the CpG uptake? Are they directly

inhibiting the nucleic acid-sensing TLRs?

- TLR3 and TLR7 have also been involved in rheumatoid arthritis. Experiments exploring the effect of the cNP after transfecting for instance TLR3 ligand are needed. Can circulating endogenous dsRNA be measured in the in vivo experiments?
- Why did the authors use lipofectamine to stimulate SFMC and FLS with cfDNA? Is this cfDNA able to stimulate these cells without being transfected?
- Why do the authors think that rats died more after injection with PDMA, than with PLGA-PDMA NPs?
- please, histological quantification (H&E and safranin) is needed to complete the in vivo studies.

Dear Editor,

We would like to thank the reviewers for their constructive comments. Please find our detailed response in a point-to-point manner as listed below. Corresponding changes have been made in the manuscript and Supplementary information as indicated. We look forward to your favourable responses soon.

Reviewer #1 (Remarks to the Author):

1. The binding of the cNP with cfDNA is mainly through charge interaction. It is well known that highly charged nanoparticles may adsorb serum proteins upon injection resulting in neutralization of the surface charge. It is recommended that the authors should demonstrate the DNA binding of the cNP in the presence or serum or serum albumin to confirm that the protein coating would minimally impact the cNP-cfDNA interaction as well as other biological functions of the cNP.

Reply: The point is well taken. Actually, DNA binding efficiency in the first submission was determined in 10% FBS. In the revision (Figure 1b), we updated the data in PBS for comparison. We confirmed that the binding efficiency was not affected with or without serum at polymer to DNA ratio greater than 0.6. Only in low ratio (<0.6), the binding efficiency of cationic materials in the presence of serum was lower than that in PBS. Efficacy of DNA scavenging was demonstrated in vivo, of which the level of cfDNA in systemic circulation was reduced to the background level in CIA rats. We have revised the text in paragraph 1 on Page 4.

2. Quantitative results for the biodistributions of all major organs are desired based on the images shown in Figure 6a.

Reply: Thank you for the suggestion. In the revision, we have quantified the images and presented the results in Supplementary Figure 8b and c. The quantitative results are also briefly discussed in the revised manuscript (paragraph 2 of page 12).

3. Histology analysis all major organs other than lung (in Supplementary Figure 8.) should be provided to further confirm the toxicity profile.

Reply: Previously we had presented the acute toxicity of cationic polymers in lung and the long-term toxicity in liver and kidneys as reflected by the levels of ALP, ALT, AST, creatinine, urea, and uric acid, as shown in Supplementary Figure 8 (in original submission). According to the suggestion, we have included the histological analysis of the heart, liver, spleen, lung, and kidney for long-term toxicity. These data are added and reorganized in Supplementary Figure 10 and 11 (revised manuscript). Corresponding alteration was made in the text as shown in paragraph 2 on Page 16.

4. The chemical structure of PLGA in supplementary document should be more accurate. It is a random copolymer so an “r” should be written between the repeating units.

Reply: Corrected as suggested. Thank you.

5. Figure S6: Although it is not the focus of the figure, it should be acknowledged that the NIR fluorescence from major organs may not be observed due to the presence of the fur.

Reply: Thanks for the suggestion. We have included this statement in the figure caption of Supplementary Figure 8a.

Reviewer #2 (Remarks to the Author):

- Can you please mention in the introduction the link between cell-free DNA and NETs, that has also been linked to rheumatoid arthritis?

Reply: Thank you for the suggestion. We have added the following text and references to the Introduction (see paragraph 1 on page 2): The origin of cfDNA comes from the degradation of DNA (or DNA fragments) released from dying or dead cells, nuclei expelled from erythroid precursors, mitochondrial DNA, and neutrophil extracellular traps (NETs)^{1, 2}.

Although elevated levels of cfDNA have been correlated with RA, the inflammatory effects of cfDNA on RA have not been reported previously. This study suggests this relationship. NETs, on the other hand, have been linked to RA. NETs are released from NETosis, which is a specialized form of neutrophil death leading to the rupture of neutrophil cell membrane. The nuclear material (DNA and histones) as well as granule enzymes (myeloperoxidase, elastase, lactoferrin, matrix metalloproteinase-9) and cytoplasmic proteins (LL37) would be released. Previous studies have focused on the protein components of NETs as the main source of autoantigens that induce autoimmune diseases including RA³. The effect of released dsDNA from NETs on inflammation has not been reported before. The cationic materials reported in this study would also likely interact with NETs and these interactions would warrant further studies.

- Please elaborate the rationale to use cNP instead of other options studied in previous studies (Lee et al, 2011). If the main novelty of this manuscript is the improved material used in this study, studies that compare both materials are needed.

Reply: The main novelty of this work rests with the use of cNP as nucleic acid (NA) scavenger for RA. The previous work has focused on soluble polycations as NA scavengers. We hypothesized that cNP may have unique scavenging properties due to its different extracellular and intracellular distribution characteristics from soluble polycations. For instance, cNP trapped in the reticuloendothelial system may scavenge cfDNA in the circulation more efficiently - indeed a point shown in this study. The longer retention time or higher AUC of cNP also helps. Moreover, the cNP may accumulate in the inflamed joint once reaching there, which appears to be the case in the imaging study. The cNP will also be sequestered in the endosomal compartment and inhibit the TLR-9 pathway activation more efficiently than soluble polycations, a

point demonstrated in this study. To properly compare the effects of cNP and soluble polycations, we synthesized the PLGA-*b*-PDMA because PDMA by itself is a potent polycation used in nonviral gene delivery, and PLGA provides the hydrophobic core to form a cNP. PLGA in addition offers biodegradability and low toxicity. The reason to choose PDMA is because it may be prepared by controlled polymerization. Therefore, the main novelty of this work is to establish that cationic nanostructured materials can modulate inflammation in an unexpected manner and suggests a new direction for nanomedicine. Heeding the suggestion of the reviewer, we have added part of this explanation to the Discussion section (see paragraph 1 on page 14) to clarify the rationale and emphasize the novelty of this work.

- Also for the readers no familiar with NP, could you please elaborate why these materials bind DNA? Can they bind to something else? Meaning can they work as scavengers of other negatively charged pro-inflammatory molecules released in the extracellular space from dying cells?

Reply: Thank you for the suggestion, and the question is one of the most important as well as challenging issues for this scavenging approach. The binding of NA to cNP stems from charge interaction. The cNP should bind also negative charged proteins. However, the interaction with NA should be stronger than that with proteins. Therefore, in the presence of serum, cNP may still bind cfDNA efficiently as shown in this work (Figure 1b). We believe that the immune complexes between cfDNA and anti-DNA antibody may be destroyed by cNP. However, the question on whether other negatively charged pro-inflammatory molecules have been removed is one of the most important and fundamental issues of this new approach of modulating inflammation. We are devoting effort to answer this question, which will require extensive effort and it is beyond the scope of this manuscript.

- More complete studies using different TLRs reporters are needed to show that the new material is only inhibiting nucleic acid sensing TLRs (TLR3, TLR7 and TLR9 vs TLR2 or TLR4 or other stimuli like TNF/IL1). Also experiments and Raw and FLS using different ligands including mammalian DNA and the cNP are needed to suggest TLR specificity.

Reply: We appreciate the suggestion. We have performed new experiments and added the new data in Supplementary Figure 2 and discussed them in the revised manuscript (see the last paragraph on page 4). We found that cNP and PDMA inhibited the activation of Ramos blueTM reporter cells and RAW264.7 cells by poly (I:C) and CpG, but not Pam3CSK4 and R848, which is consistent with previous reports⁴. In addition, we studied the inhibitory effects of the cationic materials on the stimulation of ssRNA40, which is an agonist for mouse TLR7, on RAW264.7 cells. However, we could not study their effects on the Ramos BlueTM reporter cells, which are derived from human B lymphocytes and cannot sense ssRNA40.

Unfortunately, FLS is not a good candidate for studying the TLR specificity because

the action of TLR in FLS is still unclear. Ospelt et al showed that the stimulation of FLS with ligands for TLRs 7, 8, and 9 did not elicit IL-6⁵. Although FLS could express TLR9, Kyburz et al demonstrated that CpG ODN could not activate TLR signaling of FLS⁶. Here, we tried to stimulate FLS with TLR7 and TLR9 agonists such as CpG ODN 2006, ssRNA40, and R848, but could not detect any effect. Therefore we did not include these data concerning TLR specificity in the revision. However, we believe this study on using cfDNA obtained from the synovial fluid of patients to stimulate primary SFMC and FLS should have addressed that concern, and it should be a more meaningful way of comparing the mammalian DNA stimulation in RAW and FLS.

- The authors stated that cNP inhibits intracellular agonists to TLR9. Further experiments using for instance chloroquine, that is an inhibitor of endosomal acidification, are needed to prove that cNP do not neutralize extracellular inflammatory nucleic acids. Is this the mechanism, when you add the cNP and CpG at the same time? Is the cNP also decreasing the CpG uptake? Are they directly inhibiting the nucleic acid-sensing TLRs?

Reply: We have now performed the suggested experiments, and the results are shown in Supplementary Figure 2c and Figure 4 in the Supplemental file. Chloroquine is known to disrupt the endosomal vesicles to promote escape of polyplexes and lipoplexes to the cytoplasm. Being toxic, it would also confound the experimental design and interpretation. Nevertheless, the new data in Supplementary Figure 4 reinforced the original conclusion that cNP inhibits the extracellular agonists to TLR9. When cationic polymers and CpG were added to the cells at the same time, the cationic polymers could decrease the CpG uptake. The similar reduction of CpG uptake by cationic polymers has been reported in the previous study⁴, and was briefly discussed in the revised manuscript (see paragraph 3 on page 5). Besides, to determine whether the cationic materials might bind to the TLR7 receptor and exert the inhibitory effect, we treated RAW264.7 cells with ssRNA40 and the cationic materials first and then added R848 one hour later. The new data in Supplementary Figure 2c showed that although the TLR7 stimulation would be blocked because of the binding of ssRNA40 by the cationic materials in the first step, the significant up-regulation of the inflammatory cytokines stimulated by R848 could still be observed, demonstrating that the cationic polymers neutralized the extracellular pro-inflammatory nucleic acids rather than directly inhibiting the nucleic acid-sensing TLRs. This result is also consistent with the literature^{4, 7}. We have added a brief discussion in the revised manuscript (see the last paragraph on page 4).

- TLR3 and TLR7 have also being involved in rheumatoid arthritis. Experiments exploring the effect of the cNP after transfecting for instance TLR3 ligand are needed. Can circulating endogenous dsRNA be measured in the in vivo experiments?

Reply: The point is well taken and we have performed the additional experiment as suggested. The new data described above show that the cNP could inhibit the TLR3 and TLR7 activation. However, we have focused on cfDNA in this study because its

high concentration in RA patients in both the serum and synovial fluid has been documented. The concentration of dsRNA in the serum of RA patients (70–1006 pg/mL) and in the synovial fluid (0–2093 pg/mL) is much lower than the cfDNA concentration (typically 6.79 ± 11.9 ng/mL in the serum and 5135.4 ± 11148.6 ng/mL in the synovial fluid)^{8,9}. We have found that the dsRNA concentration in the serum of normal and model rat, and synovial fluid of normal and model rat was 106, 103, 158, and 192 pg/mL, respectively, which was also very low. Since we could not detect any statistical analysis in the dsRNA concentrations of these different rat groups, we mainly focused on cfDNA rather than dsRNA.

- Why did the authors use lipofectamine to stimulate SFMC and FLS with cfDNA? Is this cfDNA able to stimulate these cells without being transfected?

Reply: cfDNA may stimulate cells only in very high concentration (>10 μ M), which is far higher than the actual concentration in vivo. Therefore, we applied Lipofectamine to deliver cfDNA into the cells for stimulation.

- Why do the authors think that rats died more after injection with PDMA, than with PLGA-PDMA NPs?

Reply: This is not an unusual finding as indicated by the literature on nonviral gene delivery, that branched and in particular star-shaped and dendritic polymers are often less toxic than their linear counterparts¹⁰. As showed in our experiment, the soluble PDMA has a different biodistribution relatively to cNP, accumulating in the lung quickly to cause acute death. This result has been reported in the literature, too¹¹. The difference in toxicity is also understandable on a cationic charge-per-mass basis; the IC50 of the cNP is nearly 2 fold higher than the soluble PDMA.

- please, histological quantification (H&E and safranin) is needed to complete the in vivo studies.

Reply: We appreciate the suggestion, and have quantified the histological evaluation in the revised manuscript (Supplementary Figure 6e and 7e). We have also added a brief discussion in the revised manuscript (see paragraph 1 on page 8 and paragraph 3 on page 10).

References

1. Jahr, S. *et al.* DNA fragments in the blood plasma of cancer patients: Quantitations and evidence for their origin from apoptotic and necrotic cells. *Cancer Res.* **61**, 1659-1665 (2001).
2. Breitbach, S., Tug, S., Simon, P. Circulating Cell-Free DNA An Up-Coming Molecular Marker in Exercise Physiology. *Sports Med.* **42**, 565-586 (2012).
3. Khandpur, R. *et al.* NETs Are a Source of Citrullinated Autoantigens and Stimulate

Inflammatory Responses in Rheumatoid Arthritis. *Sci. Transl. Med.* **5**, 10 (2013).

4. Lee, J. *et al.* Nucleic acid-binding polymers as anti-inflammatory agents. *Proc. Natl. Acad. Sci. USA* **108**, 14055-14060 (2011).
5. Ospelt, C. *et al.* Overexpression of Toll-like Receptors 3 and 4 in Synovial Tissue From Patients With Early Rheumatoid Arthritis Toll-like Receptor Expression in Early and Longstanding Arthritis. *Arthritis Rheum.* **58**, 3684-3692 (2008).
6. Kyburz, D. *et al.* Bacterial peptidoglycans but not CpG oligodeoxynucleotides activate synovial fibroblasts by toll - like receptor signaling. *Arthritis & Rheumatism* **48**, 642-650 (2010).
7. Holl, E. K. *et al.* Scavenging nucleic acid debris to combat autoimmunity and infectious disease. *Proc. Natl. Acad. Sci. USA* **113**, 9728-9733 (2016).
8. Bokarewa, M. *et al.* Arthritogenic dsRNA is present in synovial fluid from rheumatoid arthritis patients with an erosive disease course. *Eur. J. Immunol.* **38**, 3237-3244 (2008).
9. Hashimoto, T. *et al.* Circulating cell free DNA: a marker to predict the therapeutic response for biological DMARDs in rheumatoid arthritis. *Int. J. Rheum. Dis.* **20**, 722-730 (2017).
10. Synatschke, C. V. *et al.* Influence of Polymer Architecture and Molecular Weight of Poly(2-(dimethylamino)ethyl methacrylate) Polycations on Transfection Efficiency and Cell Viability in Gene Delivery. *Biomacromolecules* **12**, 4247-4255 (2011).
11. Verbaan, F. J. *et al.* The fate of poly(2-dimethyl amino ethyl)methacrylate-based polyplexes after intravenous administration. *Int. J. Pharm.* **214**, 99-101 (2001).

REVIEWERS' COMMENTS:

Reviewer #2 (Remarks to the Author):

The authors responded to all raised points convincingly. However, I would appreciate if authors could include in the discussion some of their clarifications for the readers, specifically:

"The cationic materials reported in this study would also likely interact with NETs and these interactions would warrant further studies"

"The binding of NA to cNP stems from charge interaction. The cNP should bind also negative charged proteins. However, the interaction with NA should be stronger than that with proteins. Therefore, in the presence of serum, cNP may still bind cfDNA efficiently as shown in this work (Figure 1b). We believe that the immune complexes between cfDNA and anti-DNA antibody may be destroyed by cNP. However, the question on whether other negatively charged pro-inflammatory molecules have been removed is one of the most important and fundamental issues of this new approach of modulating inflammation"

"The new data described above show that the cNP could inhibit the TLR3 and TLR7 activation. However, we have focused on cfDNA in this study because its high concentration in RA patients in both the serum and synovial fluid has been documented. The concentration of dsRNA in the serum of RA patients (70–1006 pg /mL) and in the synovial fluid (0–2093 pg/mL) is much lower than the cfDNA concentration (typically 6.79 ± 11.9 ng/mL in the serum and 5135.4 ± 11148.6 ng/mL in the synovial fluid)^{8, 9}. We have found that the dsRNA concentration in the serum of normal and model rat, and synovial fluid of normal and model rat was 106, 103, 158, and 192 pg/mL, respectively, which was also very low. Since we could not detect any statistical analysis in the dsRNA concentrations of these different rat groups, we mainly focused on cfDNA rather than dsRNA"

Other than that, the study is ready for publication in my opinion.

Dear Editor,

We would like to thank the reviewers for their constructive comments. Please find our detailed response in a point-to-point manner as listed below. Corresponding changes have been made in the manuscript and Supplementary information as indicated. We look forward to your favourable responses soon.

Reviewer #1 (Remarks to the Author):

1. The binding of the cNP with cfDNA is mainly through charge interaction. It is well known that highly charged nanoparticles may adsorb serum proteins upon injection resulting in neutralization of the surface charge. It is recommended that the authors should demonstrate the DNA binding of the cNP in the presence or serum or serum albumin to confirm that the protein coating would minimally impact the cNP-cfDNA interaction as well as other biological functions of the cNP.

Reply: The point is well taken. Actually, DNA binding efficiency in the first submission was determined in 10% FBS. In the revision (Figure 1b), we updated the data in PBS for comparison. We confirmed that the binding efficiency was not affected with or without serum at polymer to DNA ratio greater than 0.6. Only in low ratio (<0.6), the binding efficiency of cationic materials in the presence of serum was lower than that in PBS. Efficacy of DNA scavenging was demonstrated in vivo, of which the level of cfDNA in systemic circulation was reduced to the background level in CIA rats. We have revised the text in paragraph 1 on Page 4.

2. Quantitative results for the biodistributions of all major organs are desired based on the images shown in Figure 6a.

Reply: Thank you for the suggestion. In the revision, we have quantified the images and presented the results in Supplementary Figure 9b and c. The quantitative results are also briefly discussed in the revised manuscript (paragraph 2 of page 12).

3. Histology analysis all major organs other than lung (in Supplementary Figure 8.) should be provided to further confirm the toxicity profile.

Reply: Previously we had presented the acute toxicity of cationic polymers in lung and the long-term toxicity in liver and kidneys as reflected by the levels of ALP, ALT, AST, creatinine, urea, and uric acid, as shown in Supplementary Figure 8 (in original submission). According to the suggestion, we have included the histological analysis of the heart, liver, spleen, lung, and kidney for long-term toxicity. These data are added and reorganized in Supplementary Figure 11 and 12 (revised manuscript). Corresponding alteration was made in the text as shown in paragraph 2 on Page 16.

4. The chemical structure of PLGA in supplementary document should be more accurate. It is a random copolymer so an “r” should be written between the repeating units.

Reply: Corrected as suggested. Thank you.

5. Figure S6: Although it is not the focus of the figure, it should be acknowledged that the NIR fluorescence from major organs may not be observed due to the presence of the fur.

Reply: Thanks for the suggestion. We have included this statement in the figure caption of Supplementary Figure 9a.

Reviewer #2 (Remarks to the Author):

- Can you please mention in the introduction the link between cell-free DNA and NETs, that has also been linked to rheumatoid arthritis?

Reply: Thank you for the suggestion. We have added the following text and references to the Introduction (see paragraph 1 on page 2): The origin of cfDNA comes from the degradation of DNA (or DNA fragments) released from dying or dead cells, nuclei expelled from erythroid precursors, mitochondrial DNA, and neutrophil extracellular traps (NETs)^{1, 2}.

Although elevated levels of cfDNA have been correlated with RA, the inflammatory effects of cfDNA on RA have not been reported previously. This study suggests this relationship. NETs, on the other hand, have been linked to RA. NETs are released from NETosis, which is a specialized form of neutrophil death leading to the rupture of neutrophil cell membrane. The nuclear material (DNA and histones) as well as granule enzymes (myeloperoxidase, elastase, lactoferrin, matrix metalloproteinase-9) and cytoplasmic proteins (LL37) would be released. Previous studies have focused on the protein components of NETs as the main source of autoantigens that induce autoimmune diseases including RA³. The effect of released dsDNA from NETs on inflammation has not been reported before. The cationic materials reported in this study would also likely interact with NETs and these interactions would warrant further studies. We have revised the text in paragraph 2 of the Discussion section.

- Please elaborate the rationale to use cNP instead of other options studied in previous studies (Lee et al, 2011). If the main novelty of this manuscript is the improved material used in this study, studies that compare both materials are needed.

Reply: The main novelty of this work rests with the use of cNP as nucleic acid (NA) scavenger for RA. The previous work has focused on soluble polycations as NA scavengers. We hypothesized that cNP may have unique scavenging properties due to its different extracellular and intracellular distribution characteristics from soluble polycations. For instance, cNP trapped in the reticuloendothelial system may scavenge cfDNA in the circulation more efficiently - indeed a point shown in this study. The longer retention time or higher AUC of cNP also helps. Moreover, the cNP may accumulate in the inflamed joint once reaching there, which appears to be the case in the imaging study. The cNP will also be sequestered in the endosomal compartment

and inhibit the TLR-9 pathway activation more efficiently than soluble polycations, a point demonstrated in this study. To properly compare the effects of cNP and soluble polycations, we synthesized the PLGA-*b*-PDMA because PDMA by itself is a potent polycation used in nonviral gene delivery, and PLGA provides the hydrophobic core to form a cNP. PLGA in addition offers biodegradability and low toxicity. The reason to choose PDMA is because it may be prepared by controlled polymerization. Therefore, the main novelty of this work is to establish that cationic nanostructured materials can modulate inflammation in an unexpected manner and suggests a new direction for nanomedicine. Heeding the suggestion of the reviewer, we have added part of this explanation to the Discussion section (see paragraph 1 on page 14) to clarify the rationale and emphasize the novelty of this work.

- Also for the readers no familiar with NP, could you please elaborate why these materials bind DNA? Can they bind to something else? Meaning can they work as scavengers of other negatively charged pro-inflammatory molecules released in the extracellular space from dying cells?

Reply: Thank you for the suggestion, and the question is one of the most important as well as challenging issues for this scavenging approach. The binding of NA to cNP stems from charge interaction. The cNP should bind also negative charged proteins. However, the interaction with NA should be stronger than that with proteins. Therefore, in the presence of serum, cNP may still bind cfDNA efficiently as shown in this work (Figure 1b). We believe that the immune complexes between cfDNA and anti-DNA antibody may be destroyed by cNP. However, the question on whether other negatively charged pro-inflammatory molecules have been removed is one of the most important and fundamental issues of this new approach of modulating inflammation. We are devoting effort to answer this question, which will require extensive effort and it is beyond the scope of this manuscript. We have revised the text in paragraph 2 of the Discussion section.

- More complete studies using different TLRs reporters are needed to show that the new material is only inhibiting nucleic acid sensing TLRs (TLR3, TRL7 and TLR9 vs TLR2 or TLR4 or other stimuli like TNF/IL1). Also experiments and Raw and FLS using different ligands including mammalian DNA and the cNP are needed to suggest TLR specificity.

Reply: We appreciate the suggestion. We have performed new experiments and added the new data in Supplementary Figure 3 and discussed them in the revised manuscript (see the last paragraph on page 4). We found that cNP and PDMA inhibited the activation of Ramos blueTM reporter cells and RAW264.7 cells by poly (I:C) and CpG, but not Pam3CSK4 and R848, which is consistent with previous reports⁴. In addition, we studied the inhibitory effects of the cationic materials on the stimulation of ssRNA40, which is an agonist for mouse TLR7, on RAW264.7 cells. However, we could not study their effects on the Ramos BlueTM reporter cells, which are derived

from human B lymphocytes and cannot sense ssRNA40.

Unfortunately, FLS is not a good candidate for studying the TLR specificity because the action of TLR in FLS is still unclear. Ospelt et al showed that the stimulation of FLS with ligands for TLRs 7, 8, and 9 did not elicit IL-6⁵. Although FLS could express TLR9, Kyburz et al demonstrated that CpG ODN could not activate TLR signaling of FLS⁶. Here, we tried to stimulate FLS with TLR7 and TLR9 agonists such as CpG ODN 2006, ssRNA40, and R848, but could not detect any effect. Therefore we did not include these data concerning TLR specificity in the revision. However, we believe this study on using cfDNA obtained from the synovial fluid of patients to stimulate primary SFMC and FLS should have addressed that concern, and it should be a more meaningful way of comparing the mammalian DNA stimulation in RAW and FLS.

- The authors stated that cNP inhibits intracellular agonists to TLR9. Further experiments using for instance chloroquine, that is an inhibitor of endosomal acidification, are needed to prove that cNP do not neutralize extracellular inflammatory nucleic acids. Is this the mechanism, when you add the cNP and CpG at the same time? Is the cNP also decreasing the CpG uptake? Are they directly inhibiting the nucleic acid-sensing TLRs?

Reply: We have now performed the suggested experiments, and the results are shown in Supplementary Figure 3c and Figure 5 in the Supplemental file. Chloroquine is known to disrupt the endosomal vesicles to promote escape of polyplexes and lipoplexes to the cytoplasm. Being toxic, it would also confound the experimental design and interpretation. Nevertheless, the new data in Supplementary Figure 5 reinforced the original conclusion that cNP inhibits the extracellular agonists to TLR9. When cationic polymers and CpG were added to the cells at the same time, the cationic polymers could decrease the CpG uptake. The similar reduction of CpG uptake by cationic polymers has been reported in the previous study⁴, and was briefly discussed in the revised manuscript (see paragraph 3 on page 5). Besides, to determine whether the cationic materials might bind to the TLR7 receptor and exert the inhibitory effect, we treated RAW264.7 cells with ssRNA40 and the cationic materials first and then added R848 one hour later. The new data in Supplementary Figure 3c showed that although the TLR7 stimulation would be blocked because of the binding of ssRNA40 by the cationic materials in the first step, the significant up-regulation of the inflammatory cytokines stimulated by R848 could still be observed, demonstrating that the cationic polymers neutralized the extracellular pro-inflammatory nucleic acids rather than directly inhibiting the nucleic acid-sensing TLRs. This result is also consistent with the literature^{4, 7}. We have added a brief discussion in the revised manuscript (see the last paragraph on page 4).

- TLR3 and TLR7 have also being involved in rheumatoid arthritis. Experiments exploring the effect of the cNP after transfecting for instance TLR3 ligand are needed. Can circulating endogenous dsRNA be measured in the in vivo experiments?

Reply: The point is well taken and we have performed the additional experiment as suggested. The new data described above show that the cNP could inhibit the TLR3 and TLR7 activation. However, we have focused on cfDNA in this study because its high concentration in RA patients in both the serum and synovial fluid has been documented. The concentration of dsRNA in the serum of RA patients (70–1006 pg/mL) and in the synovial fluid (0–2093 pg/mL) is much lower than the cfDNA concentration (typically 6.79 ± 11.9 ng/mL in the serum and 5135.4 ± 11148.6 ng/mL in the synovial fluid)^{8,9}. We have found that the dsRNA concentration in the serum of normal and model rat, and synovial fluid of normal and model rat was 106, 103, 158, and 192 pg/mL, respectively, which was also very low. Since we could not detect any statistical analysis in the dsRNA concentrations of these different rat groups, we mainly focused on cfDNA rather than dsRNA. We have revised the text in paragraph 3 of the Discussion section.

- Why did the authors use lipofectamine to stimulate SFMC and FLS with cfDNA? Is this cfDNA able to stimulate these cells without being transfected?

Reply: cfDNA may stimulate cells only in very high concentration (>10 μ M), which is far higher than the actual concentration in vivo. Therefore, we applied Lipofectamine to deliver cfDNA into the cells for stimulation.

- Why do the authors think that rats died more after injection with PDMA, than with PLGA-PDMA NPs?

Reply: This is not an unusual finding as indicated by the literature on nonviral gene delivery, that branched and in particular star-shaped and dendritic polymers are often less toxic than their linear counterparts¹⁰. As showed in our experiment, the soluble PDMA has a different biodistribution relatively to cNP, accumulating in the lung quickly to cause acute death. This result has been reported in the literature, too¹¹. The difference in toxicity is also understandable on a cationic charge-per-mass basis; the IC50 of the cNP is nearly 2 fold higher than the soluble PDMA.

- please, histological quantification (H&E and safranin) is needed to complete the in vivo studies.

Reply: We appreciate the suggestion, and have quantified the histological evaluation in the revised manuscript (Supplementary Figure 7e and 8e). We have also added a brief discussion in the revised manuscript (see paragraph 1 on page 8 and paragraph 3 on page 10).

References

1. Jahr, S. *et al.* DNA fragments in the blood plasma of cancer patients: Quantitations and evidence for their origin from apoptotic and necrotic cells. *Cancer Res.* **61**, 1659-1665 (2001).

2. Breitbach, S., Tug, S., Simon, P. Circulating Cell-Free DNA An Up-Coming Molecular Marker in Exercise Physiology. *Sports Med.* **42**, 565-586 (2012).
3. Khandpur, R. *et al.* NETs Are a Source of Citrullinated Autoantigens and Stimulate Inflammatory Responses in Rheumatoid Arthritis. *Sci. Transl. Med.* **5**, 10 (2013).
4. Lee, J. *et al.* Nucleic acid-binding polymers as anti-inflammatory agents. *Proc. Natl. Acad. Sci. USA* **108**, 14055-14060 (2011).
5. Ospelt, C. *et al.* Overexpression of Toll-like Receptors 3 and 4 in Synovial Tissue From Patients With Early Rheumatoid Arthritis Toll-like Receptor Expression in Early and Longstanding Arthritis. *Arthritis Rheum.* **58**, 3684-3692 (2008).
6. Kyburz, D. *et al.* Bacterial peptidoglycans but not CpG oligodeoxynucleotides activate synovial fibroblasts by toll - like receptor signaling. *Arthritis & Rheumatism* **48**, 642-650 (2010).
7. Holl, E. K. *et al.* Scavenging nucleic acid debris to combat autoimmunity and infectious disease. *Proc. Natl. Acad. Sci. USA* **113**, 9728-9733 (2016).
8. Bokarewa, M. *et al.* Arthritogenic dsRNA is present in synovial fluid from rheumatoid arthritis patients with an erosive disease course. *Eur. J. Immunol.* **38**, 3237-3244 (2008).
9. Hashimoto, T. *et al.* Circulating cell free DNA: a marker to predict the therapeutic response for biological DMARDs in rheumatoid arthritis. *Int. J. Rheum. Dis.* **20**, 722-730 (2017).
10. Synatschke, C. V. *et al.* Influence of Polymer Architecture and Molecular Weight of Poly(2-(dimethylamino)ethyl methacrylate) Polycations on Transfection Efficiency and Cell Viability in Gene Delivery. *Biomacromolecules* **12**, 4247-4255 (2011).
11. Verbaan, F. J. *et al.* The fate of poly(2-dimethyl amino ethyl)methacrylate-based polyplexes after intravenous administration. *Int. J. Pharm.* **214**, 99-101 (2001).